



# Rain-fed streams dilute inorganic nutrients but subsidise organic matter-associated nutrients in coastal waters of the northeast Pacific Ocean

Kyra A. St. Pierre[1,2], Brian P.V. Hunt[1,2,3], Suzanne E. Tank[1,4], Ian Giesbrecht[1,5], Maartje C. Korver[1,6],
William C. Floyd[7,8], Allison A. Oliver[1,4,9], Ken P. Lertzman[1,5]

[1]Hakai Institute, Tula Foundation, Heriot Bay BC, V0P 1H0, Canada
[2]Institute for the Oceans and Fisheries, University of British Columbia, Vancouver BC, V6T 1Z4, Canada
[3]Department of Earth, Ocean and Atmospheric Sciences, University of British Columbia, Vancouver BC, V6T 1Z4, Canada
[4]Department of Biological Sciences, University of Alberta, Edmonton AB, T6G 2E9, Canada
[5]School of Resource and Environmental Management, Simon Fraser University, Burnaby BC, V5A 1S6, Canada
[6]*Current address*: McGill University, Department of Geography, Montréal QC, H3A 0B9, Canada
[7]Ministry of Forests, Lands and Natural Resource Operations, Nanaimo BC, V9T 6E9, Canada
[8]Vancouver Island University, Nanaimo BC, V9R 5S5, Canada
[9]*Current address*: Skeena Fisheries Commission, Kispiox BC, V0J 1Y4 Canada

*Correspondence to*: Kyra A. St. Pierre (k.stpierre@oceans.ubc.ca)

**Abstract.** In coastal regions, rivers and streams may be important sources of nutrients limiting to primary production in marine waters; however, sampling is still only rarely conducted across the land-to-ocean aquatic continuum, precluding conclusions from being drawn about connectivity between freshwater and marine systems. Here we use a more than four year dataset (2014-2018) of nutrients (nitrogen, phosphorus, silica, iron) and dissolved organic carbon spanning streams draining coastal watersheds and nearshore marine surface waters along the Central Coast of British Columbia Canada, at the heart of the North Pacific coastal temperate rainforest region. Mean freshwater and surface marine N:Si:P ratios were 4:18:1 (P:Fe = 1:58) and 6:11:1, respectively, showing relative consistency across the land-ocean interface but deviation from the extended Redfield ratio. Inorganic nutrient concentrations ($NO_3^-+NO_2^-$, $PO_4^{3-}$, $Si(OH)_4$) in freshwaters were less than in the receiving marine environment, indicating that freshwater nutrient inputs in this region were of little importance to - or even diluted - the pool of readily available inorganic nutrients in nearshore waters. Conversely, freshwaters increased the pool of organic matter-associated nutrients, namely dissolved organic nitrogen and iron. The organic matter-rich landscapes of the region yielded globally significant quantities of dissolved organic nitrogen (304-381 kg km$^{-2}$ y$^{-1}$) and iron (463-596 kg km$^{-2}$ y$^{-1}$); thus acting as important sources of potentially limiting nutrients to both nearshore and offshore waters. These exports may subsidise heterotrophic microbial communities capable of directly consuming and remineralising these nutrients, potentially compensating for the dilution of inorganic nutrients by freshwater inputs. We highlight the need to better understand nutrient limitation in coastal waters and for concerted research efforts to study the spatial and temporal dynamism at the land-ocean interface along the northeast Pacific coast.



## 1 Introduction

All living organisms require a combination of more than 20 naturally occurring chemical elements (Sterner and Elser, 2002).
Whereas early studies were solely focused on carbon (C), nitrogen (N) and phosphorus (P), the fundamental building blocks
of life (Redfield, 1934), the crucial role of micronutrients like silicon (Si, hereafter as orthosilicic acid, $Si(OH)_4$) and iron (Fe)
has long been recognised (Ho et al., 2003). Primary producers like phytoplankton, whose carbon uptake capabilities are
intrinsic to the coastal carbon cycle (Bauer et al., 2013), must obtain these essential nutrients directly from their environment,
and are thus dependent on the sources and sinks that govern nutrient speciation and availability.

Alfred Redfield established that marine phytoplankton are generally composed of C, N and P in a ratio of 106:16:1 (Redfield,
1934), a global average that has been shown to vary over space (Martiny et al., 2013). Updated iterations of the ratio include
Si and Fe (C:N:Si:P:Fe = 106:16:15:1:0.1-0.01) among other elements, in recognition of the important role that these elements
play in fundamental cellular metabolism and growth (Sterner and Elser, 2002; Ho et al., 2003), with special reference to
diatoms in the case of Si (Brzezinski, 1985). We now understand that nutrient limitation to primary production is determined
by biological demand, available supply, and supply relative to that of other essential elements (Welti et al., 2017).

One of the potential sources of limiting nutrients in marine waters, especially in coastal environments, is subsidies from
terrestrial ecosystems on adjacent landmasses (Sakamaki and Richardson, 2008; Harding and Reynolds, 2014; Moore et al.,
2013). As rivers and streams flow from their headwaters to coastal ecosystems, they integrate processes happening across their
watersheds, transporting the products of erosion, weathering, biological production and decomposition, all sources of nutrients
potentially limiting to primary production in the marine environment (Bouwman et al., 2013). The land-to-ocean aquatic
continuum (LOAC) concept acknowledges that what happens on land has important consequences for water quality and
productivity in coastal regions (Xenopoulos et al., 2017). However, the LOAC has typically been applied either by freshwater
scientists to describe changes in water quality across watersheds, leading up to but not including coastal waters (e.g., Bouwman
et al., 2013; Weyhenmeyer and Conley, 2017), or by marine scientists to infer freshwater inputs from salinity gradients across
nearshore environments (e.e.g, Wetz et al., 2006; Cuevas et al., 2019). Each of these approaches, while valuable, relies on
inference about the neighbouring ecosystems, thus overlooking potential local deviations from average or assumed conditions.

The North Pacific Coastal Temperate Rainforest (NPCTR) region of North America, which extends from northern California
to the Gulf of Alaska (Fig. 1; Alaback, 1996), receives upwards of 2000 mm of rainfall per year (DellaSala, 2011). High
rainfall, combined with snow and glacial melt from high elevations, translate to intense stream and river discharge (Royer
1982; Neal et al. 2010; Morrison et al. 2012), making the connection between land and sea across this region particularly

strong (Fleming et al., 2016; O'Neel et al., 2015). Indeed, along the rainfall-dominated outer coast of British Columbia, the quality of marine dissolved organic carbon (DOC) in ocean surface waters very closely resembles that of freshwater DOC during the winter months when high rainfall-induced inputs of DOC may favour microbial heterotrophy (St. Pierre and Oliver et al., 2020). While the large organic carbon stocks in soils across the NPCTR (McNicol et al., 2019) have made DOC a major focus of study (Fellman et al., 2009a; Fellman et al., 2009b; Oliver et al., 2017; Hood et al., 2006; and others), we do not fully

understand how these high freshwater exports impact nutrient availability and limitation in nearshore waters across the region (but see for example Wetz et al. 2006), a key oversight given the strong association between nutrients and organic matter for potentially limiting elements like nitrogen and iron (Meybeck, 1982).

Since 2014, routine monthly water quality surveys have been conducted in both coastal marine and fresh waters across Calvert

and Hecate Islands at the mid-latitudes of the NPCTR along the Central Coast of British Columbia, Canada (51.7°N, 128.0°W; Fig. 1) (Giesbrecht et al. 2016 and in review). To address the issues highlighted above, the objectives of this study were to: 1) understand seasonal variability in inorganic and organic nutrient concentrations and stoichiometry in nearshore waters, 2) quantify and characterise temporal variability in freshwater nutrient inputs to marine waters, and 3) assess whether freshwaters constitute an important source of potentially limiting nutrients to marine ecosystems in this region. This more than four-year

dataset offers insights into the spatial and temporal variability in nutrient stoichiometry across the land-ocean aquatic continuum, highlighting the possible role of small rivers and streams in regulating biological productivity in nearshore waters of the NPCTR.

## 2 Methods

### 2.1 Site descriptions

Kwakshua and Meay Channels separate Calvert and Hecate Islands within the Hakai Luxvbalis Conservancy along the Central Coast of British Columbia, Canada (Fig. 1). Within the channels, surface waters (0–5 m) are typically quite fresh (< 30 salinity units) relative to the rest of the water column (~30-33 salinity units). The channel system is sheltered from direct offshore influences by Calvert Island, but exchanges waters with Fitz Hugh Sound to the east through Kwakshua Channel ($26.8 \text{ km}^2$),

and with Hakai Pass and Queen Charlotte Sound to the northwest through Meay Channel ($8.6 \text{ km}^2$). Pruth Bay ($3.1 \text{ km}^2$) connects Meay and Kwakshua Channels. We hereafter refer to the combined areas of Pruth Bay, and Meay and Kwakshua Channels as the Kwakshua Channel system. Thirteen marine stations were sampled and located throughout the Kwakshua Channel system: six were approximately mid-channel, and seven were closer to the shoreline and associated with freshwater outlets (Fig. 1, Table S1).




Calvert and Hecate Islands are characterised by bog forests and extensive wetlands, and are located within the hypermaritime zone of the perhumid NPCTR (Meidinger and Pojar, 1991; Thompson et al., 2016). Mean annual 1981-2010 air temperature and precipitation near sea level on the islands were 8.93±0.20 °C and 2800±49 mm, respectively (climatewna.com, Wang et al. 2016). Precipitation on Calvert and Hecate Islands is primarily in the form of rain; however, above 500 m elevation, seasonal

snow packs over 2 m deep can develop, corresponding to 600 to 1000 mm of stored water. Snow in watersheds with significant areas of elevation greater than 500 m can contribute to a muted spring freshet, though all elevations have the potential to contribute to rain-on-snow events through late fall and early winter. The geology underlying the region is acidic plutonic, dominated by silicate and aluminium oxides, and iron oxides towards the eastern coasts of the islands (Roddick, 1996). This geology is overlain by organic-rich podzol and folic histosol soils, with thick hemists in depressional areas, and open wetlands

and short wet-forests composed of western redcedar, yellow-cedar, shore pine and western hemlock (Oliver et al., 2017). Seven streams draining coastal watersheds of the Kwakshua Channel system have been routinely sampled for a suite of dissolved chemical species (see Table S2). These watersheds, previously described in detail in Oliver et al. (2017) and summarised in Table S3, cumulatively account for 67.4% of the terrestrial drainage area (69.6 km$^2$) of the combined channels. Briefly, these watersheds range between 3.2 and 12.8 km$^2$, with extensive but variable coverage by wetlands (23.8 to 50.2%) and lakes (0.3

to 9.1%), and mean slopes between 21.7 and 40.3% (Gonzalez Arriola et al., 2015; Oliver et al., 2017).

### 2.1.1 Seasonality and wider scale climate anomalies

The NPCTR is characterised by high seasonality across both marine and freshwater systems. The spring transition occurs between March and May each year (Thomson et al., 2014), with a shift towards northwesterly winds favouring the propagation of nutrient-rich upwelled waters onto the shelf and into inner passage waters, including Kwakshua Channel (Hunt et al., 2018).

Increased solar radiation favours a phytoplankton bloom coinciding with the spring transition and high productivity through to approximately August. Between September and November, downwelling-favourable southeasterly winds begin to dominate. This change overlaps with a marked increase in rainfall and therefore terrestrial runoff to the marine system, which persists from around October through to March (St. Pierre and Oliver et al., 2020).

The wider northeast Pacific region is also characterised by large scale climate cycles, including the Pacific Decadal Oscillation (PDO) and the El Niño Southern Oscillation (ENSO), both of which can have a large effect on temperature (air, surface water) and precipitation patterns across the region (Kiffney et al., 2002). Variability in the PDO occurs over 10+ year time scales and is thus outside the scope of the present study. The study period was characterised by large fluctuations in the ENSO: a prolonged and strong El Niño from Nov-2014 to May-2016, weak La Niñas from Aug-2016 to Dec-2016 and Oct-2017 to

Mar-2017, and a weak El Niño from Oct-2018 through to the end of the record (31-Dec-2018).

**2.2 Sampling scheme and sample collection**

Field surveys of both marine and fresh waters were conducted on an approximately monthly basis between 1-Aug-2014 and 31-Dec-2018 (4 years, 5 months) by staff at the Hakai Institute's Calvert Island Field Station. Interannual variability over this period is discussed in section 3.1. Briefly, 13 marine stations and seven freshwater streams were sampled to provide
information on spatial and temporal variability in marine waters within the context of variable freshwater inputs.

**2.2.1 Marine stations in Kwakshua and Meay Channels**

Marine water samples were collected from a boat at discrete depths (0, 1 and 5 m) within the upper water column using a Niskin water sampler across the 13 marine stations. Samples for combined nitrate-nitrite ($NO_3^-+NO_2^-$; hereafter shortened to $NO_3^-$), orthophosphate ($PO_4^{3-}$), silicic acid ($Si(OH)_4$), and dissolved organic carbon (DOC) were filtered through 0.45 μm
Millipore® Millex-HP hydrophilic polyethyl sulfonate (PES) filters in the field and kept on ice until returning to the field station. Following sample collection on the boat, the basic properties (temperature, salinity, conductivity) of the collected water were measured using a calibrated, marine sample-specific YSI in a pre-rinsed container.

**2.2.2 Rainforest streams**

In this study, we integrated the results of two freshwater sampling programs conducted between 1-Aug-2014 and 31-Dec-
2018: 1) routine monthly samplings of the seven watersheds at their outlets to Kwakshua Channel, and 2) targeted rainfall event samplings of the streams over the rising and falling limbs of the hydrographs.

Discharge, rainfall and air temperature were monitored continuously at each of the seven watersheds. Climate stations were established adjacent to the seven outlet streams at low elevation (range 10-90 m.a.s.l; Table S3), near the stream sampling
location. Air temperature and precipitation were recorded at 5-minute intervals and aggregated to daily statistics for temperature (mean, minimum, maximum) or totals for precipitation. Discharge was continuously measured at 5-minute intervals, but the dataset was filtered to 15-minute intervals for flux model construction, as in Oliver et al. (2017) and St. Pierre and Oliver et al. (2020). Rating curve construction and discharge calculation have been previously described in detail in Oliver et al. (2017) and updated in Korver et al. (2019a). Weather data from Mt. Buxton (~1000 m.a.s.l.), the highest point on Calvert
Island, were also extracted for the period of record from the ClimateNA spatially downscaled model (climatewna.com; Wang et al., 2016) for comparison to the sea level stations.

Standardised monthly temperature and precipitation anomalies were calculated relative to the 1981-2010 climate normal for each station, extracted from the ClimateNA model (Wang et al., 2016). We assessed the validity of this approach by comparing
monthly ClimateNA outputs with local observations for the study period (January 2015 – December 2018). Mean percent difference between measured and modelled monthly mean air temperatures and total precipitation over the period of record



were 8.82±0.87% and 11.58±0.50%, respectively. The model predicted mean monthly air temperatures well (linear regression: slope = 0.93, $R^2$ = 0.98; Fig. S1), but tended to under predict rainfall (slope = 0.70, $R^2$ = 0.76; Fig. S2). However, rainfall is a notoriously difficult climate parameter to measure accurately, let alone model and will improve over time with the incorporation of additional data sources. Given the fit statistics, we deemed the Climate NA model adequate for our purposes. Together, these data support the use of a model-derived climate normal for the region in the absence of 30 years of local climate measurements. Local temperature and precipitation anomalies were then compared to the Oceanic Niño Index (ONI, Climate Prediction Center, 2019) to assess the potential influence of Pacific Ocean-basin scale climate anomalies on local processes.

Bulk water samples were collected as grab samples from each of the seven streams. Samples for $NO_3^-$, $PO_4^{3-}$, $Si(OH)_4$, and DOC were sampled as for the marine stations. At all freshwater stations, waters were also filtered with 0.45 µm PES filters for ammonium ($NH_4^+$), total dissolved nitrogen (TDN), total dissolved phosphorus (TDP), and dissolved iron (dFe). Additional bulk water samples were collected in 50 mL centrifuge tubes for total nitrogen (TN) and total phosphorus (TP) and kept cool until analysis. These additional parameters are discussed separately from $NO_3^-$, $Si(OH)_4$, $PO_4^{3-}$ and DOC, in acknowledgement that analogous measurements were not taken in marine waters, except for during the rainfall events (see below).

### 2.2.3 Targeted stream samplings during rainfall events

The streams were sampled during rainfall events to better understand how water chemistry changes along the rising and falling limbs of the hydrograph during storms (Korver et al., 2019b). These high frequency measurements are especially useful to train constituent flux models during high flow events given that constituent concentrations often vary with discharge within a given watershed (Goñi et al., 2013).

During rainfall events, the streams were sampled using a combination of techniques: 1) opportunistic grab samples on the falling limb (all watersheds, 18-Jul-2015 to 15-Nov-2018); 2) using an automated rack sampler (all watersheds, 21 to 25-July-2015 and 15 to 16-Jul-2017); and, 3) using an automated pump sampler (watershed 708 only, 17 to 27-Sept-2015 and 14 to 21-Oct-2017). The rack samplers were constructed by mounting 250-mL bottles at vertical increments above the low water level of the stream prior to each rain event. Vertical increments were adjusted depending on the weather forecast and predicted stage rise and as the water level rose, the bottles filled in sequence. Water exchange after filling was prevented by following the design of single-stage suspended-sediment samplers (InterAgency Committee on Water Resources, 1961): two inverted stainless-steel U-shaped inlet tubes were inserted in the bottle through a rubber plug. The first inlet tube allows water to enter and the second inlet tube, placed directly above the first, blocks water inflow by sealing the bottle's air exchange. Bottles were always completely filled at rising water levels, preventing river water from entering as water levels drop again. Additional information on the setup is included in the Supplementary Information (see also Table S4).



### 2.2.4 Freshwater plume samplings in nearshore waters

To better understand the influence of rainfall events on nearshore environments, the plumes emanating from watersheds 819 and 703 were sampled during storms on 7-Aug-2015 and 19-Sept-2015, respectively (Fig. 1). Plume extent was first visually determined based on turbidity and water colour, and finally by measuring the conductivity of surface waters in comparison to the Pruth marine station (~45,000 µS cm$^{-1}$). Six sampling stations were then established across the plume and waters were sampled at 0, 1 and 5 m depth for all the same parameters as the streams, described above.

### 2.3 Sample preparation and chemical analyses

TP and TDP samples were preserved with 80 µL of 95% $H_2SO_4$ and frozen until analysis. Samples for TN, TDN, $NH_4^+$, $NO_3^-$, and $Si(OH)_4$ were frozen until analysis. Samples for TN, TDN, $NH_4^+$, $NO_3^-$ (from Aug. 2014 to June 2015), TP, TDP, and $PO_4^{3-}$ (from Aug. 2014 to June 2015) were analysed according to Canadian Association of Laboratory Accreditation (CALA)-certified protocols at the University of Alberta's Biogeochemical Analytical Service Laboratory (Table S2). From March 2015

onwards, $NO_3^-$, $PO_4^{3-}$ and $Si(OH)_4$ were analysed at the University of British Columbia's Marine Zooplankton and Micronekton Laboratory. Dissolved inorganic nitrogen (DIN) and dissolved organic nitrogen (DON) were calculated as the sum of $NH_4^+$ and $NO_3^-$, and the difference between TDN and DIN, respectively. Dissolved cations, including dFe, were preserved with 480 µL 8M $HNO_3$, and kept cool until analysis by the Analytical Chemistry Services Laboratory (Ministry of Environment and Climate Change Strategy, Victoria, BC, Canada). Filtered DOC samples from freshwater and marine sites were preserved with

200 µL 7.5M $H_3PO_4$ or 6 M HCl, respectively, and kept cool until analysis. Freshwater samples were analysed at Analytical Chemistry Services Laboratory, while marine samples were analysed at the Ján Veizer Stable Isotope Laboratory (University of Ottawa, ON, Canada). Instruments and detection limits for each constituent are summarised in Table S2.

### 2.4 Freshwater biogeochemical inputs to Kwakshua and Meay Channels

   Freshwater inputs to the Kwakshua Channel system were quantified by relating constituent concentrations from both the

routine monthly and targeted rainfall event samplings to 15-minute discharge using log-linear models in the R package *rloadest* (Runkel et al., 2004; Runkel, 2013; Lorenz et al., 2015). For all watersheds except 708, models were constructed by relating discharge to concentration over the entire study period. At watershed 708, discharge monitoring began in Aug. 2013 - one year earlier than elsewhere, so models were constructed for the entire period of record (9-Aug-2013 to 31-Dec-2018). Daily flux estimates were, however, only made over the same period as for the other watersheds (1-Aug-2014 to 31-Dec-2018). Half of

the detection limit (see Table S2) was used for cases where concentrations of $NO_3^-$, $NH_4^+$, and $PO_4^{3-}$ were flagged as below detection by the analytical labs.

   For each constituent and watershed, the best *rloadest* model was initially chosen to minimise Akaike's Information Criterion (AIC), and then assessed for goodness-of-fit using the bias percentage, partial load ratio (PLR; Stenback et al., 2011), and the





Nash-Sutcliffe Efficiency Index ($E$; Nash and Sutcliffe, 1970). Only the models for $NO_3^-$ consistently had an $E$ less than zero, indicating that the observed mean was a better predictor than the model estimates (Runkel et al., 2004; Lorenz et al., 2015; Runkel, 2013), and were therefore excluded from our analysis. Instead, the $NO_3^-$ flux was estimated by subtracting the $NH_4^+$ flux from the DIN flux on a given day. Mean ($\pm$ SE, $n = 77$) model $R^2$, bias percentage, PLR and $E$ were 94.7 $\pm$ 0.6, -0.76 $\pm$ 0.85%, 0.99$\pm$0.01, and 0.78$\pm$0.03, respectively. Constituent and watershed-specific model fit statistics are presented in Table S5. Errors reported on aggregate monthly or annual flux measurements were quantified by propagating the daily standard error

of prediction, which incorporates the uncertainty of both the model fit and the load estimate.

**2.5 Other statistical considerations and analyses**

All statistical analyses were completed in R (R Core Team, 2019), using packages *vegan* (Oksanen et al., 2018), *dplyr* (Wickham et al., 2019), *lme* (Bates et al., 2015) and *lsmeans* (Lenth, 2016). Stoichiometric ratios were log-transformed prior

to the calculation of summary statistics presented in figures in acknowledgement of the inherent non-normality of ratio data (Isles, 2020). Mean concentrations presented in figures and tables were then re-transformed to non-log scale to facilitate comparisons with literature values. Standard errors (SE) are reported throughout, unless otherwise stated. The level of significance ($\alpha$) was 0.05, but Bonferroni-corrected where multiple comparisons were made. Statistics quantifying interannual variability utilise full calendar years only ($n = 4$, i.e., 2015-2018). Temperature and precipitation anomalies were compared to

the ONI using Pearson's product moment correlation.

Differences in air temperature, rainfall, flow-weighted freshwater nutrient concentrations and fluxes, and marine nutrient concentrations across the period of record were assessed using linear mixed effects models, coding for year and month as fixed effects and watershed or station (marine) as a random effect. We chose to compare monthly flow-weighted nutrient

concentrations, rather than point measurements to assess "true" temporal differences in concentration, independently of hydrology. Briefly, monthly flow-weighted nutrient concentrations were calculated by dividing the modelled monthly flux by the total monthly volume of water discharged by each stream. Tukey multiple comparisons were then assessed across all pairwise combinations of year and month; however, only interannual comparisons for a given month (e.g., July 2015 vs. July 2018) were deemed relevant for our purposes and are discussed.


Principal component analysis (PCA) was used to characterise the bulk chemistry of the freshwater inputs to the Kwakshua channel system. Prior to performing the PCA, flow-weighted molar concentrations of all chemical constituents were standardised using Z-scores, so as to de-emphasise the very high DOC concentrations, which were typically two to three orders of magnitude higher than any other constituent. Each principal component (PC) was then interpreted as a bulk indicator of a

particular aspect of the freshwater chemistry and used to examine changes in chemical composition over time.





## 3 Results

### 3.1 Temperature and precipitation between 2014 and 2018

Mean annual air temperature near sea-level on Calvert and Hecate Islands over the study period was 8.73±0.12 °C (full range: -13.60-34.99 °C), similar to the 1981-2010 normal for these watershed outlet sites (8.93±0.20 °C). Mean annual rainfall was

2790±61 mm (range: 2240 to 3520 mm; 1981-2010 normal = 2800±49 mm) at these sites, increasing to 4312±106 mm at 1000 m elevation (1981-2010 normal = 4620 mm). Precipitation translated to a cumulative mean annual freshwater discharge to Kwakshua Channel of 0.174±0.055 km$^3$ (Table 1, S3, range: 0.161-0.186 km$^3$), equivalent to a mean annual specific runoff of 2500±790 mm (range: 2310-2670 mm).

Rain (≥ 0.2 mm) was recorded at sea level between 63 and 69% of days in any given year (Table S6), and the region exhibited strong seasonality. Maximum and minimum monthly rainfall typically occurred in November and June, respectively (Fig. 2a). The longest (21-47 consecutive days with more than 0.2 mm of rain) and largest (> 200 mm of rain, maximum = 567 mm event$^{-1}$) rainfall events always occurred between November and March (Table S6). Maximum and minimum air and sea-water temperatures occurred in June and December-February (Fig. 2b-c). The sampling period was characterised by dramatic

fluctuations in the ONI, including strong El Niño (May-2015 to Apr-2016), and weak La Niña (Oct-2017 to Mar-2016) and El Niño (Oct-2018 to Dec-2018) events (Fig. 2d). Temperature and precipitation anomalies, though, were poorly correlated with the ONI (Fig. S3).

### 3.2 Marine primary production and nutrient stoichiometry

Mean monthly chlorophyll $a$ concentrations ranged between 0.06±0.01 µg L$^{-1}$ (Nov. 2014) and 4.38±1.71 µg L$^{-1}$ (Jun. 2016;

Fig. 3a). Each year, primary production peaked in March-April and again in June-July, based on chlorophyll $a$ concentrations (Fig. 3a). Mean marine NO$_3^-$, PO$_4^{3-}$ and Si(OH)$_4$ concentrations across the 0, 1, and 5 m depths were 8.18, 0.78, and 18.67 µmol L$^{-1}$, respectively, but varied widely (Table S7). NO$_3^-$, PO$_4^{3-}$ and Si(OH)$_4$ concentrations displayed strong seasonality, but also interannual differences (Table 2; mixed effects models, p < 0.05 on all terms, Table S8). Seasonal fluctuations in inorganic nutrient concentrations in marine waters were the opposite of chlorophyll $a$ concentrations: the depletion of NO$_3^-$, PO$_4^{3-}$ and

Si(OH)$_4$ occurred annually in concert with the peak of primary production, followed by increases in concentration through the late summer and autumn as chlorophyll $a$ concentrations decreased (Fig. 3b-d). There was a 37-fold increase in NO$_3^-$ concentrations between the monthly minimum (June mean: 0.57±0.16 µmol L$^{-1}$) and maximum (December mean: 21.23±0.57 µmol L$^{-1}$). The same seasonal increase was 10-fold for PO$_4^{3-}$ (July: 0.16±0.04 µmol L$^{-1}$; December: 1.57±0.01 µmol L$^{-1}$) and 12-fold for Si(OH)$_4$ (June: 2.74±0.47 µmol L$^{-1}$; December: 38.29±0.37 µmol L$^{-1}$). Mean DOC concentrations remained low

throughout the year (72.66±2.64 µmol L$^{-1}$; Fig. 3e).





Mean $NO_3^-$:$Si(OH)_4$, $NO_3^-$: $PO_4^{3-}$, and $Si(OH)_4$: $PO_4^{3-}$ ratios were 0.3:1, 6:1, and 11:1 respectively. All ratios shifted seasonally (Fig. 4). $NO_3^-$:$Si(OH)_4$ and $NO_3^-$: $PO_4^{3-}$ reached a minimum in the summer (June-August) and increased again through the autumn and winter (September through February). $NO_3^-$: $Si(OH)_4$ was consistently below the Redfield-Brzezinski ratio for
diatoms (16:15 = 0.94), oscillating between < 0.1 between May and July and ~0.6 in January (Fig. 4a), and indicating an excess of $Si(OH)_4$. $NO_3^-$: $PO_4^{3-}$ consistently declined to ~1:1 by July of each year, and approached the Redfield ratio (16:1) during the winter months (Fig. 4b). $Si(OH)_4$: $PO_4^{3-}$ ratios were close to the Redfield ratio (15:1) or above it (Fig. 4c), with less of a discernible seasonal trend than for the other ratios. Greater variability between the stations was typically observed in the spring and summer months.

## 3.3 Freshwater nutrient exports to nearshore waters of the NPCTR

### 3.3.1 Freshwater nutrient and DOC concentrations

Mean measured concentrations of $NO_3^-$, $PO_4^{3-}$, $Si(OH)_4$ and DOC in the streams were 0.47, 0.09, 1.67, and 954 µmol $L^{-1}$, respectively and showed strong seasonal and spatial variability (Table S7). On average, measured concentrations of $NO_3^-$, $PO_4^{3-}$ and $Si(OH)_4$ were 48±21 (1.4-720), 9.87±1.20 (0.62-28.15), and 12.42±1.43 (0.25-28.6) times lower in freshwater than across
the marine stations (Fig. 3b-d). The mean monthly concentrations of $PO_4^{3-}$ and $Si(OH)_4$ in freshwaters only exceeded those in marine waters on one occasion (July 2018). Seasonal variability in freshwater inorganic nutrient concentrations was muted, compared to marine waters (Fig. 3b-d). In contrast, freshwater concentrations of DOC were 14.55±0.99 (6.98-36.29) times higher than in marine waters and exhibited high seasonal variability, increasing by almost 2-fold between the low in February (641±69.8 µmol $L^{-1}$) and the high in August (1150±51.3 µmol $L^{-1}$; Fig. 3e).


In general, flow-weighted inorganic nutrient concentrations were less than the measured concentrations whereas flow-weighted organic nutrient concentrations were higher than the measured concentrations, reflecting the dependence of concentration on flow within the watersheds (Table S7). Although flow-weighted concentrations of DOC fluctuated seasonally (Fig. S4), this seasonal cycle was consistent between years (mixed effect model: only month term p-value < 0.05). $PO_4^{3-}$ concentrations were
approximately constant throughout the year (p-value on all terms > 0.05), while $NO_3^-$ and $Si(OH)_4$ concentrations were variable across the period of record (year-month interaction p < 0.05) (Fig. S4).

Summary statistics for measured and flow-weighted concentrations of TN, TDN, DON, DIN, $NH_4^+$, TP, TDP, and dFe are reported in Table S7. Flow-weighted concentrations of TN, TDN, DON, and dFe varied seasonally (linear mixed effects
models: month term p-value<0.05; Table S9). As for $PO_4^{3-}$, TP and TDP concentrations were approximately constant throughout the year. DIN and $NH_4^+$ concentrations were variable across the period of record (year-month interaction p<0.05). DIN concentrations were driven largely by a dramatic increase in concentration in August 2018 associated with $NO_3^-$. Trends in $NH_4^+$ concentrations were also highly variable, with consistent interannual differences from August through December in





2015 (monthly means: 0.778–0.827 µmol L$^{-1}$) and 2018 (0.277–0.387 µmol L$^{-1}$). We note, however, that concentrations of the

inorganic N species and all P species were often near or below detection.

### 3.3.2 Freshwater nutrient fluxes, speciation and stoichiometry

DOC was by far the largest terrestrial input from the seven watersheds to Kwakshua Channel (mean = 143±20.4 Mmol yr$^{-1}$; 128 to 156 Mmol yr$^{-1}$; Table 1), exceeding all other inputs by two to three orders of magnitude. TN fluxes (2.10±0.30 Mmol yr$^{-1}$) were overwhelmingly DON (81.8±15.5%), with the remainder as DIN (8.8±4.3%) or PN. Around half (~53.8%) of DIN

was exported as $NH_4^+$, and the other half as $NO_3^-$. TP exports were extremely low (0.07±0.02 Mmol yr$^{-1}$), only 18.9±11.0% of which was as the readily available form $PO_4^{3-}$, with the balance presumably split between particulate phosphorus (~11%, PP = TP–TDP) and dissolved organic phosphorus (~63%, DOP = TP–PP–$PO_4^{3-}$).

Mean $NO_3^-$:$Si(OH)_4$, $NO_3^-$: $PO_4^{3-}$, and $Si(OH)_4$: $PO_4^{3-}$ ratios for freshwater fluxes were 0.20, 3.69, and 18.03, respectively.

$NO_3^-$:$Si(OH)_4$ and $NO_3^-$: $PO_4^{3-}$ ratios were consistently at or below the Redfield ratio (Fig. 4). Notably, $NO_3^-$:$PO_4^{3-}$ increased in the late summer. Stoichiometric ratios were, however, highly variable between the different watersheds (Fig. S5). Clear seasonal signals were difficult to discern across most watersheds, except for watersheds 703 and 708. Watershed 703 in particular displayed late summer peaks across all ratios. Fe:$PO_4^{3-}$ far exceeded the extended Redfield ratio (0.1-0.01:1) across all watersheds (57.51; Fig. S5d), and fluctuated synchronously across most watersheds, driven by the late summer increases

in Fe concentrations (Fig. S4).

### 3.3.3 Freshwater yields of nutrients and DOC in a global context

TN yields (378-463 kg km$^{-2}$ y$^{-1}$) were at the low end of the global range (1–20,630 kg km$^{-2}$ y$^{-1}$), but within the range for coniferous forests (Alvarez-Cobelas et al., 2008) (Table S10). Meanwhile, DON yields (304-381 kg km$^{-2}$ y$^{-1}$) were at the high end of the estimated global range (10–479 kg km$^{-2}$ y$^{-1}$; Alvarez-Cobelas et al., 2008), and exceeded by up to 6-times other

estimated and modelled mean global and North American yields (Table S10). In contrast, DIN yields (33.6-40.9 kg km$^{-2}$ y$^{-1}$) were at the low end of the global ranges for $NO_3^-$ (Alvarez-Cobelas et al. 2008), but nearly identical to the mean value reported by Meybeck (1982). $Si(OH)_4$ yields from the watersheds (84.5-95.7 kg km$^{-2}$ y$^{-1}$) were up to 26-times lower than the reported North American average (Dürr et al., 2011). Dissolved Fe yields (463-596 kg km$^{-2}$ y$^{-1}$) exceeded the estimated global riverine yield by between 55- and 710-times, depending on the estimate (De Baar and De Jong, 2001; Krachler et al., 2005). As

previously described for a slightly different study period (Oliver et al., 2017), DOC yields (22,200–26,900 kg km$^{-2}$ y$^{-1}$) from the Kwakshua Channel drainage basin exceeded by almost two times the upper bound of the global range (2000–14,000 kg km$^{-2}$ y$^{-1}$) reported by Meybeck (1982), and exceeded by up to 18 and 26-times the modelled global and North American yields, respectively (Seitzinger et al., 2005) (Table S10), although our combined estimate was lower than that reported for the 2015 water year in Oliver et al. (2017) (33,300 kg km$^{-2}$ yr$^{-1}$).



### 3.3.4 Organic matter-associated versus inorganic nutrients

Measured TN, TDN, DON, and dFe concentrations pooled across all watersheds were positively correlated with DOC (Pearson product moment correlation, $r = 0.612$-$0.738$, depending on the species; Table S11). Conversely, concentrations of DIN, $NH_4^+$, $NO_3^-$, TP, TDP, $PO_4^{3-}$, and $Si(OH)_4$ were only weakly positively ($r < 0.300$ for TP, TDP, $PO_4^{3-}$, $NH_4^+$, $Si(OH)_4$) or negatively ($-0.300 < r < 0.000$ for DIN, $NO_3^-$) associated with DOC. Based on these relationships, we distinguish hereafter between strongly organic matter-associated (DON – which makes up most of the TDN and TN pools – and dFe) and inorganic (DIN, $PO_4^{3-}$, $Si(OH)_4$) nutrients in the freshwater pool. The strength and direction of these relationships were, however, highly watershed-specific (Table S11). This was especially true for $Si(OH)_4$, for which watershed-specific correlation coefficients ranged between -0.15 and 0.60. In particular, there was a strong positive correlation between concentrations of $Si(OH)_4$ and DOC from watersheds 708 and 693 (Table S11), indicating the likely association with organic matter in those watersheds.

### 3.3.5 Patterns in bulk freshwater chemistry

We used PCA to describe the bulk freshwater chemistry independent of discharge over time using monthly flow-weighted DOC and nutrient concentrations. Two PCs alone accounted for 63.8% of total observed variability. PC1, which accounted for 50.0% of total variability, was defined by TDN, TN, Fe, DON, DOC, $Si(OH)_4$, and TP, suggesting that it represented primarily dissolved organic matter (DOM)-associated compounds (Fig. 5a, Table S12). PC1 showed strong spatial separation between watersheds and seasonal variability over the period of record (Fig. 5b). Watersheds with the highest yields of DOC (626, 819, 844; Oliver et al., 2017) also exported high concentrations of the DOM-associated suite of nutrients (TDN, TN, Fe, DON). Across most watersheds, loadings on PC1 were highest in June-July, indicating higher concentrations of DOM-associated compounds, reaching a minimum in February-March. There was low interannual variability in PC1, suggesting a consistent seasonal export of DOM-associated nutrients from freshwater ecosystems.

PC2 (13.8%) was negatively associated with TDP, $NH_4^+$, TP and DIN, but positively with $PO_4^{3-}$ and $Si(OH)_4$, and thus represented the inorganic nutrients (Fig. 5a). Like PC1, PC2 oscillated seasonally, but was much more consistent between watersheds with the exception of watershed 708 (Fig. 5c; see Table S3 for watershed information). This pattern is reflective of the observed increases in concentrations of DIN, $NH_4^+$ and TDP in the late winter/early spring, whereas concentrations of $PO_4^{3-}$ and $Si(OH)_4$ increased in late summer/early autumn. Unlike PC1, there was some interannual variability in PC2.

### 3.4 Mixing in nearshore waters

Plume concentrations of $NO_3^-$, $PO_4^{3-}$, and $Si(OH)_4$ were consistently higher than the freshwater reference concentrations during both rainfall events (Fig. 6, S6). Conversely, freshwater concentrations of DOC, DON and dFe were much higher than in marine waters. During the August rainfall event (Fig. 6), all of the surveyed nutrients (except for DOC) mixed non-conservatively, with $NO_3^-$, $PO_4^{3-}$, and dFe mixing below conservative behaviour, and $Si(OH)_4$ and DON mixing above it.

During the September 2015 event (Fig. S6), $NO_3^-$, $PO_4^3$ and DOC mixed approximately conservatively across most of the plume, whereas $Si(OH)_4$ concentrations were well above conservative mixing, suggesting an additional source of $Si(OH)_4$ to surface waters during the rain events. DON and dFe were below conservative mixing.

During both events, $NO_3^-$: $Si(OH)_4$ was below the Redfield ratio across the whole plume (Fig. 7, S7). $NO_3^-$: $PO_4^{3-}$ was above the Redfield ratio at lower salinities (5–10 salinity units) but approached the ratio across the rest of the plume during the September event (Fig. S7), and was well below the Redfield ratio across the whole plume except for one station in August (Fig. 7). Fe: $PO_4^{3-}$ and Fe: $NO_3^-$ were close to the Redfield ratio across the plumes during both events.

### 3.5 Role of inorganic freshwater nutrient inputs in subsidising nearshore productivity

By scaling estimates of primary productivity measured at nearby Rivers Inlet (51.7°N, 127.3°W; Shiller, 2012) to the surface area of Kwakshua Channel, we previously estimated that total primary production within the Kwakshua Channel system is on the order of 21 to 42 Gg C yr$^{-1}$ (St. Pierre and Oliver et al., 2020). Assuming that phytoplankton nutrient requirements are at the extended Redfield ratio, freshwater DIN and $Si(OH)_4$ inputs could, at most, directly support primary production on the order of 0.02 Gg C yr$^{-1}$, or less than 1% of the estimated total. Freshwater $PO_4^{3-}$ fluxes likewise could only support up to 0.02

Gg C yr$^{-1}$ (based on the largest annual fluxes from 2016; Table 1). Assuming no loss, the freshwater dFe inputs could support between 7.3 Gg C yr$^{-1}$ (based on P:Fe = 0.1) and 94.5 Gg C yr$^{-1}$ (based on P:Fe = 0.01), or between 17.4% and 450% of the estimated primary production.

## 4 Discussion

We conducted routine monthly and targeted rainfall event surveys, linking marine-terminating streams and nearshore surface waters from Aug-2014 through to Dec-2018 to quantify the flux of terrestrial materials to nearshore ecosystems from small bog-forest watersheds with a hypermaritime climate. Below we discuss spatial and temporal variability of these fluxes and nutrient availability in receiving nearshore ecosystems, the consequences of terrestrial exports for nearshore ecosystem function, and highlight priority research areas (Fig. 8).

**4.1 Decoupling of small hypermaritime watershed exports from wider scale climate anomalies**

The relationship between short term weather patterns along the northeast Pacific coast and the ONI has been well described, with El Niño events associated with warmer air temperatures and lower rainfall/stream flow, and La Niña events corresponding to colder and wetter periods (Ward et al., 2010). We did not, however, find this to be true at the scale of the Kwakshua Channel system, where air temperature and precipitation anomalies were unrelated to the ONI (Fig. S3). Whereas most work relating

climate anomalies to stream flow has focused on large river basins (Wang et al., 2006) or continental scales (Ward et al., 2010),





our results suggest that localised freshwater inputs from smaller hypermaritime watersheds of the NPCTR may be a more consistent input to nearshore surface waters. The small size of the watersheds limits retention of precipitation, such that rain events directly translate into enhanced stream flows. That being said, ENSO events may have resulted in more intense storms or longer periods of low flow, a temporal scale which was not specifically examined here. In the marine environment, El Niño

and La Niña events have been associated with changes in nutrient availability in the northeast Pacific (Whitney and Welch, 2002). Intensification of stratification during El Niño events can lead to nutrient depletion in the surface ocean during the summer productive season (Whitney and Welch, 2002). Conversely, during La Niña events deeper mixing can favour higher nutrient concentrations in surface waters (Whitney and Welch, 2002).

## 4.2 Importance of freshwater exports for nearshore primary production

### 4.2.1 Origin of nutrients in freshwater systems of the NPCTR

Nutrients in freshwater ecosystems of the NPCTR originate from a diverse array of potential sources, including soils and terrestrial ecosystems, interactions with the atmosphere, and the return of migratory fish species (Sugai and Burrell, 1984; Hood et al., 2007; Fellman et al., 2009c). Inorganic nitrogen species in freshwaters may originate from the fixation of atmospheric $N_2$ or the ammonification of organic matter to $NH_4^+$, with subsequent nitrification to $NO_3^-$ (Wetzel, 2001). On the

other hand, P has a geologic source and is mobilised by mineral weathering, which releases $PO_4^{3-}$, among other ions (Walker and Syers, 1976). However, the quartz diorite bedrock in the area is poor in P, with less than 0.2% of the mineral content accounted for by P-containing compounds (Roddick, 1996). The mineral P pool from which non-occluded P ($PO_4^{3-}$ sorbed to Fe and aluminium oxides; Walker and Syers, 1976) can be mobilised is thus very small, such that remineralisation of organic material is likely to be the primary source of P in these watersheds. The small size of the watersheds and the large and frequent

rainfall events characteristic of the NPCTR may limit in-watershed reprocessing of the large quantities of DOM, except in those few watersheds with larger lake and wetland areas (Oliver et al., 2017). Based on lithology (Hartmann and Moosdorf, 2012) and the hypermaritime boundary (Salkfield et al., 2016), we determined that the bedrock of Calvert and Hecate Islands (acidic plutonic) is widespread on the BC outer coast, accounting for roughly 37% (11,210 km²) of the total area of the hypermaritime region (30,080 km²). This suggests that low P fluxes are likely representative of many small coastal watersheds

located within the NPCTR.

Concentrations of DOM-associated nutrients like DON and dFe increased seasonally following the end of the drier summer period (Fig. S4). This pattern is a well-known feature of transitional periods in highly seasonal catchments, including snow-covered catchments during the spring melt (Boyer et al., 1997) and in coastal catchments with distinct dry and wet seasons,

like those sampled here (Sanderman et al., 2009; Oliver et al., 2017; Fellman et al., 2009a). When hydrological connectivity is low between soils and streams during drier periods, DOM accumulates in soils, a by-product of microbial degradation of organic matter. With the onset of the autumn rains, the accumulated DOM is then flushed into streams. This transition is also





associated with a change in the source and therefore lability of the DOM, discussed in detail for these watersheds in Oliver et al. (2017). dFe in particular may be subject to dissolution by organic acids at the end of the summer (Ling Ong et al., 1970;
Keller, 2019), and the resultant Fe-OM complexes can then be exported to streams and rivers (Jones, 1998; Boyer et al., 1997). In a similar vein, ratios involving $NO_3^-$ also consistently showed increased relative $NO_3^-$ concentrations in July and/or August in each year (Fig. 4). We surmise that this may be due to increased remineralisation of DOM coupled with the oxygenation of previously waterlogged soils during the dry summer months.

The seasonal increase in concentration of the DOM-associated nutrients was also observed for $Si(OH)_4$, albeit to a lesser extent and with high spatial variability (Fig. S4). Dissolved $Si(OH)_4$ presence in freshwater systems is largely a function of local geology. Its availability is then determined by a number of factors, including DOM concentration and quality, internal processing in lakes and streams (e.g., redox cycling, phytoplankton uptake), water temperature, and pH (Wetzel, 2001). $Si(OH)_4$ concentrations in the streams were highly variable and the origin of $Si(OH)_4$ is not obvious. When pooled across all
samples, $Si(OH)_4$ was unrelated to DOC (Table S11). For certain watersheds, though, $Si(OH)_4$ was strongly positively correlated with DOC ($r > 0.50$ for watersheds 708 and 693), suggesting that $Si(OH)_4$ there was associated with DOM. However, there was no relationship or even a weak negative one for others (Table S11). The highest concentrations of $Si(OH)_4$ were consistently observed in watersheds 703 and 626, and were unrelated to DOC, suggesting that deposits of fine sediments found within both watersheds may support greater mineral weathering (Eamer and Shugar, 2015). $Si(OH)_4$ concentrations in the
streams of Calvert and Hecate islands (0.24 to 37.89 μmol L$^{-1}$) were well below the global riverine average of 150 μmol L$^{-1}$ (Conley, 1997), and in fact more similar to the concentrations observed in marine waters (Table S7), suggesting little, if any, influence of freshwater $Si(OH)_4$ exports over the Kwakshua Channel marine ecosystem as a whole.

### 4.2.2 Role of freshwater nutrient inputs to nearshore ecosystems

Our results suggest that freshwater inputs of most inorganic nutrients (DIN, $PO_4^{3-}$) can directly support less than 1% of the
estimated primary production in nearshore surface waters in this particular area of the NPCTR. In many cases, terrestrial DOM-associated nutrients are not directly available to phytoplankton in nearshore waters, and their incorporation into food webs first requires a mineralisation step (Eppley and Peterson, 1979; Hedges et al., 1997), or cleavage by exo-enzymes (Benitez-Nelson, 2000). Our earlier work in Kwakshua Channel suggested that large inputs of DOM during rainfall events may favour microbial growth in surface waters (St. Pierre and Oliver et al., 2020), which could enhance remineralisation. Indeed, in the Baltic Sea,
increasing terrestrial inputs have been associated with a shift from largely autotrophic communities to increasing heterotrophy in surface waters (Wikner and Andersson, 2012). Studies in estuaries around the world have also highlighted the potential role of terrestrial DOM in subsidising marine zooplankton production (Hoffman et al., 2008; Hitchcock et al., 2016), though the extent to which that is the case here is unknown (Fig. 8).



The fact that concentrations of key inorganic nutrients (DIN, $PO_4^{3-}$, $Si(OH)_4$) in freshwaters feeding Kwakshua Channel were consistently lower than in adjacent marine waters is unusual (Wetz et al., 2006), albeit not unheard of along the eastern Pacific coast (Cuevas et al., 2019). Freshwater inputs may therefore act to dilute inorganic nutrient pools in marine surface waters, potentially limiting primary production there if remineralisation of OM is insufficient. To date, many of the measurements spanning the land-to-ocean aquatic continuum have been made in waters flowing through heavily human-impacted watersheds

(e.g., Baltic Sea, Mississippi River/Gulf of Mexico). Intensive agricultural, industrial and urban development in these watersheds tend to increase freshwater inorganic nutrient delivery to coastal waters (e.g., Perez et al., 2011). Our results highlight the need to study less anthropogenically disturbed regions to gain a better global understanding of nutrient and carbon dynamics at the land-ocean interface, and the degree to which the functioning of these systems has been perturbed.

Ratios of potentially limiting nutrients in marine waters consistently indicated N limitation relative to Si and P in nearshore waters (Fig. 4, 7, S7). This suggests that a) there is an additional source of nutrients – especially N – potentially unaccounted for; b) phytoplankton nutrient requirements in these waters deviate from Redfield quantities; or, c) phytoplankton experience replete growth until the limiting nutrient had been consumed. In support of the first, DON fluxes from Calvert and Hecate Islands were high, with concentrations exceeding those in marine waters during the rainfall events (Fig. 6e, S6e). DON may

be an additional source of DIN to surface waters through remineralisation and/or photochemical decomposition processes (Eppley and Peterson, 1979; Goldman et al., 1987; Tank et al., 2012). In Arctic coastal waters, for example, it is estimated that microbes incorporate at least 74.6% of the nitrogen that they consume, with DIN regeneration accounting for up to the remaining 25.4% (Tank et al., 2012). This, combined with photoammonification, may result in total DIN regeneration rates from terrestrial DON inputs of between 55 and 83% (Tank et al., 2012). By applying these rates to the freshwater DON fluxes

from Calvert and Hecate Islands, we estimate that between 0.83 and 1.57 Mmol $yr^{-1}$ DIN could originate from the terrestrial DON flux, effectively increasing the terrestrially-derived inorganic nitrogen pool by five to eight times. In the Baltic Sea, nitrogen regeneration can support between 78 and 97% of primary production in coastal waters (Klawonn et al., 2019); while in other coastal areas, nitrogen regeneration can more than (>100%) satisfy N demand by phytoplankton (Diaz and Raimbault, 2000), highlighting the importance of regeneration in coastal waters. Local estimates of nitrogen uptake and regeneration would, however, be valuable to corroborate this estimate (Fig. 8). Higher summertime phytoplankton growth in the nearshore

waters of Prince William Sound (Gulf of Alaska) is believed to be principally sustained by regenerated nitrogen (Strom et al., 2006), suggesting that some phytoplankton communities along the NPCTR coast may be particularly well-adapted to or closely associated with microbes capable of the rapid remineralisation of organic matter-associated nutrients.

Alternatively, phytoplankton nutrient requirements may deviate from Redfield quantities, such that the use of Redfield ratios is imperfect in this region. Local or regional deviations from Redfield stoichiometry in surface waters have been widely reported (Körtzinger et al.; Martiny et al., 2013). Though not accounting for Si or Fe, the C:N:P ratio of phytoplankton declines with increasing latitude, with a mean C:N:P of 78:13:1 in nutrient-rich high latitude (45°N to 65°N) waters (Martiny et al.,



2013). The mean water column N:P ($NO_3^-$:$PO_4^{3-}$) ratio in the marine surface waters along the Central Coast was 6:1, but
fluctuated annually between ~1:1 in mid-summer and the Redfield ratio - 16:1 - in the winter (Fig. 4c). In the Gulf of Alaska,
Strom et al. (2006) observed similar but less dramatic seasonal changes in $NO_3^-$:$PO_4^{3-}$ through the productive spring and early
summer months in the inner shelf waters. As in the Kwakshua Channel system, waters along the outer shelf of the Gulf of
Alaska were also characterised by an excess of $Si(OH)_4$ relative to $NO_3^-$, conditions which were associated with the
proliferation of small (< 5 µm), non-diatomaceous cells. Although we did not characterise phytoplankton communities as part
of this study, excess $Si(OH)_4$ in surface waters could indicate that diatom production may be limited by $NO_3^-$, dFe, or another
nutrient altogether.

Like DON, concentrations of dFe in freshwaters greatly exceeded those in marine surface waters, corresponding to high fluxes
of potentially limiting dFe to the coastal environment. dFe mixed well below conservative behaviour across both freshwater
plumes (Fig. 6f, S6f), suggesting either the rapid sedimentation of dFe out of surface waters, a fraction that may occur with
the flocculation of DOC (St. Pierre and Oliver et al., 2020; Herzog et al., 2020a), or the rapid uptake of dFe into food webs.
While Fe limitation has previously been documented for diatoms in nearshore coastal waters of the North Pacific (Takeda,
1998; Bruland et al., 2001), primary productivity was not explicitly measured in this study, thus limiting the direct conclusions
that can be drawn without additional data.


Fe availability to both microbes and higher trophic level organisms in coastal waters, and more broadly across the global
oceans, depends on the presence and identity of organic ligands to which Fe binds (Lauderdale et al., 2020). Terrestrial
dissolved organic matter-bound Fe, which likely predominates in these watersheds, has been shown to be highly stable during
estuarine mixing, and can be transported well beyond the estuary to open waters where it can serve as a source of potentially
limiting-Fe (Herzog et al., 2020b). However, despite this stability, these complexes remain permeable to microbial
siderophores (Batchelli et al., 2010), molecules with a strong affinity for Fe produced by microbes across a wide diversity of
nutrient regimes in the coastal oceans (Boiteau et al., 2019). The quantification of dFe complexation and more broadly of fate
(sedimentation, uptake) are important next steps within this region (Fig. 8).

### 4.2.3 Spatial and temporal variability in nearshore mixing

There are several possible reasons why the mixing of $NO_3^-$, $PO_4^{3-}$, and DON varied between the two rainfall events (Fig. 6,
S6). Firstly, the plumes sampled emanated from two different watersheds (819 on 07-August and 703 on 19-September), which
have amongst the highest and lowest organic carbon yields of the study watersheds, respectively (Oliver et al., 2017). Because
the quality and quantity of the freshwater exports differ so dramatically, some of the differences observed in mixing may
reflect differences in the freshwater end-member. The outlet stream of watershed 819 drains extensive wetlands (50.2%
watershed area; Table S3), ecosystems known to be hotspots of DOM production and processing. On the other hand, watershed
703 has the thickest mineral soils (mean depth = 35.8 cm; Oliver et al. 2017), exposed bedrock at high elevation, a steep





watershed gradient, and comparatively low coverage by lakes and wetlands, indicating little time for retention and processing of nutrients within the watershed before export to coastal waters. We note, however, that the freshwater end-members, while watershed and month-specific, were not sampled at the same time as the plume and may therefore not accurately reflect nutrient concentrations during these specific rainfall events (see also Fig. 3 for variation in the freshwater endmember). Secondly, the sampling time within each of the respective events was quite different. The sampling in August occurred on the fifth day of a six-day storm during a drier period, with rainfall of ~62 mm up to and including sampling on 07-August (total rainfall of ~85 mm over six days, previous 30-day total = ~100 mm). In contrast, the September sampling occurred on the second day of an eight-day event, with rainfall of ~49 mm up to and including sampling on 19-September (total rainfall of ~135 mm over eight days, previous 30-day total = ~190 mm). Rainfall events of these durations and sizes are among the most common in any given year (Table S6), and as such, these events were representative of medium sized events during the warmer seasons. Finally, these two events occurred at different times of year within the context of the seasonal marine cycle: one in early August when temperatures are still warm and marine waters are still relatively productive, and the second in mid-September when the transition to lower productivity and cooler/rainier weather has typically begun. For example, $NO_3^-$ exhibited non-conservative behaviour during the August event when we would expect uptake by primary producers to be higher (Fig. 6), but conservative behaviour during the September event (Fig. S6), when primary production is somewhat lower. Regardless of the reason(s), these two events highlight the variable relationship between nearshore waters and freshwater inputs over time and space.

### 4.2.4 Nutrients across the land-ocean interface in the NPCTR

While DOC has been a major focus across the NPCTR (see Fellman et al. 2009a; Hood et al. 2006; Oliver et al. 2017; and references therein), we know relatively little about nutrient stoichiometry across the land-to-ocean aquatic continuum in this region. A handful of studies documenting nutrients in either coastal waters or streams exist, without full consideration of the composition of the adjacent waters. A notable exception to this, but outside the NPCTR of North America, includes efforts in three coastal temperate rainforest watersheds along the eastern coast of Japan. In contrast to our findings, freshwater concentrations of $NO_3^-$ (18-46 µmol L$^{-1}$) and $Si(OH)_4$ (44-172 µmol L$^{-1}$) there were higher than in nearshore waters ($NO_3^-$ = 0.7-22 µmol L$^{-1}$; $Si(OH)_4$ = 10-109 µmol L$^{-1}$), such that freshwaters may have acted as sources of these nutrients to adjacent marine systems (Matsunaga et al., 1998). Marine surveys of freshwater plumes along the Oregon Coast likewise inferred that coastal watersheds were important sources of $NO_3^-$ (43-52 µmol L$^{-1}$), $PO_4^{3-}$ (4.9-5.5 µmol L$^{-1}$), $Si(OH)_4$ (171-197 µmol L$^{-1}$), and dFe (0.12 µmol L$^{-1}$) to nearshore waters (Wetz et al., 2006). However, in both Oregon and Japan, the studied watersheds include agricultural activities, large human settlements, and more productive forests with N-fixing alder, all of which tend to increase nutrient loadings to aquatic ecosystems. In both cases, these surveys were also conducted over short time periods (1-2 months), and therefore likely do not represent the full range of possible conditions.

The nutrient and DOC concentrations that we observed in the Calvert and Hecate streams were very similar to those from earlier work in the Wilson and Blossom Rivers of southeast Alaska (Sugai and Burrell, 1984). There, synchronous late





summer/early autumn spikes in DOC and Fe concentrations were attributed to the flushing of forest soils following the drier summer period. A late summer peak in $NH_4^+$ and $PO_4^{3-}$ concentrations was associated with the large salmon runs supported by the rivers (Sugai and Burrell, 1984). Although the streams of Calvert and Hecate do not support large salmonid stocks, we did observe a small increase in flow-weighted $NH_4^+$ concentrations in August in watersheds 819, 708 and 844 (Fig. S4), two of which (708 and 844) are known to have coho salmon (*Oncorhynchus kisutch*; McAdams, 2018). However, the increase in $NH_4^+$ was not reflected in other constituents that would be associated with salmon (e.g., $NO_3^-$, $PO_4^{3-}$, DOC as in Hood et al., 2007), rather suggesting enhanced remineralisation of organic matter in soils during the summer. In nearshore waters, Wilson and Blossom River $PO_4^{3-}$ mixed conservatively and there was little loss of Fe in surficial coastal waters, but the number of marine measurements taken were limited relative to the river surveys (Sugai and Burrell, 1984). Fe yields from the Calvert and Hecate watersheds were comparable to these more northerly sites (Sugai and Burrell, 1984), which also exceeded the global riverine average by two to three orders of magnitude (Table S10).

Surface waters of Hecate Strait, just north of Queen Charlotte Sound along the British Columbia coast, were similar to the Redfield ratio in both summer and winter with respect to N:P and Si:N, and exceeded the Redfield-Brzezinski Si:P ratio (Whitney et al., 2005). The Si:P ratios (21-26:1) were similar to those in the Kwakshua Channel system, but the Hecate Strait summer N:P (13:1) was much higher than the ~1:1 in Kwakshua waters (Whitney et al., 2005), again highlighting significant spatial variability within a small area of the NPCTR. Although nutrient limitation assays in coastal waters of British Columbia are sparse, the Gulf of Alaska, which borders the northern reaches of the NPCTR, is known to be Fe-limited (Martin et al., 1989), while N co-limitation has been observed across the continental shelf (Strom et al., 2006). This spatial variability in nutrient limitation status is largely a result of the freshwater input dynamics, vertical mixing events, and horizontal transport processes, e.g., onshore transport of offshore waters during downwelling (Strom et al., 2006). Eddies generated on the continental shelf in the northeast Pacific can readily transport iron and other nutrients (e.g., DON) into offshore waters (Johnson et al., 2005; Ladd et al., 2009; Cullen et al., 2009). The existence of both dFe supply and an efficient transport mechanism suggests that that the NPCTR may be a hotspot of terrestrial Fe export to offshore waters of the northeast Pacific Ocean.

The sheer diversity of both coastal ecosystems and watershed types across the NPCTR make it difficult to apply findings uniformly from one part of the coast to the wider region. Extensive marine surveys have been conducted along the Chilean coast in the South PCTR and demonstrated highly variable nearshore nutrient profiles with differing freshwater inputs (Cuevas et al., 2019). Areas receiving greater freshwater inputs were associated with lower total chlorophyll *a* concentrations, and phytoplankton communities dominated by small cells less than 20 µm (Cuevas et al., 2019). Given that seasonality plays such an important role across the region, it is imperative that more surveys be conducted year-round in recognition that freshwater and marine influences may alter the quality of nearshore waters differently throughout the year.



### 4.3 Influence of high land-ocean connectivity in autumn-winter on the spring bloom

Given that freshwater exports to nearshore ecosystems are highest in autumn and winter, we might expect exports at this time of year to play an important role in priming nearshore waters for the spring bloom. Although this wetter season coincides with

lower primary production in receiving marine ecosystems, wintertime downwelling winds can be favourable for the retention of freshwater plumes in nearshore environments (Thomson, 1981). In some eastern boundary current upwelling regions, upwelling of terrestrial Fe deposited in coastal sediments may be an important source of Fe for nearshore waters during the summer months (Johnson et al., 1999; Chase et al., 2005). Along the Oregon coast, for example, wintertime riverine inputs stay close to shore, and freshwater inputs of limiting Fe are sufficient to support significant primary production in winter, and

the entirety of the spring-summer phytoplankton bloom (Wetz et al., 2006). With the switch to upwelling favourable winds in the spring, some of this pool can then disperse and potentially support primary production over a much wider area and into offshore waters (Herzog et al., 2020b). The large freshwater dFe exports may therefore represent an important subsidy to both nearshore surface waters and waters offshore.

Also of potential importance are the freshwater exports of DON (304-381 kg km$^{-2}$ y$^{-1}$), which were at the high end of the estimated range for rivers globally (10-479 kg km$^{-2}$ y$^{-1}$; Alvarez-Cobelas et al., 2008). Although DON has traditionally been considered resistant to microbial degradation in surface waters (Voss and Hietanen, 2013), the advective flux from coastal waters can be a potentially important source of N to the open ocean. Globally, waters of the eastern Pacific have high DON concentrations (Letscher et al., 2013), which may, at least in part, be influenced by high freshwater inputs that drive a clear

nearshore (high) to offshore (low) gradient in DON concentrations in the region (Wong et al., 2002). However, the fate of the terrestrial DON in these waters requires further investigation (Fig. 8).

The freshwater plumes that we sampled during the two rainfall events were likely quickly dissipated by tide and wind activity, but because of the sheer volume of freshwater exported during the autumn and winter months, combined with downwelling-

favourable winds, we might expect freshwater plumes to persist for longer during these seasons. The Kwakshua Channel is approximately 800 km north of the Oregon study area, where Wetz et al. (2006) found high winter primary production. Kwakshua Channel thus has colder waters and, critically, receives less incident light during the winter than the Oregon shelf. Wintertime primary production is therefore likely reduced compared to the Oregon shelf, but the observation of such high winter primary production in nearby regions highlights the potential biogeochemical importance of what is traditionally an

under-sampled season across both freshwater and marine environments. Although we cannot definitively say to what extent autumn and winter freshwater inputs may contribute to the spring bloom, such high fluxes of potentially limiting nutrients (N, dFe) could play an important role during this under-studied season.

# 5 Conclusions and next steps

Using a more than four year dataset, we show that freshwater exports from small hypermaritime rainforest watersheds along

the British Columbia outer coast to nearshore surface waters may directly dilute coastal pools of inorganic nutrients, but enhance pools of organic matter-associated nutrients that require remineralisation or chelation (dFe) before being readily available to primary producers in surface waters. These freshwater exports have potentially important consequences for the function of nearshore waters, altering food web structure and energy transfer (Fig. 8). In particular, freshwater yields of DON and dFe across the watersheds were at the high end of (DON) or greatly exceeded (dFe) currently documented global ranges

of riverine nutrient yields to coastal waters.

Based on these results, a number of future priority research directions have emerged. Of primary importance will be to understand which nutrients are limiting in coastal waters of the NPCTR throughout the year (Fig. 8). Of secondary importance will be to resolve a) the nature of the freshwater dFe exports (organic vs. inorganic complexes, identity of ligands), which

ultimately determines the lability and downstream fate of this potentially critical nutrient (Herzog et al., 2020b), and b) the fate of the freshwater DON exports within the nearshore waters (i.e., extent of microbial remineralisation, photoammonification, uptake) (Fig. 8).

Coastal waters of the northeast Pacific Ocean receive nutrients from multiple terrestrial freshwater sources, including rainfall,

snow, and glacial melt (Hood and Berner, 2009; Edwards et al., 2013), as well as from multiple marine sources, including upwelling and exchange with offshore waters (Thomson, 1981). To date, much of the work examining the impacts of this watershed diversity on nearshore ecosystems has focused on differences in the lability of dissolved organic matter from these various hydrological sources to marine ecosystems (e.g., Arimitsu et al., 2018; Hood and Berner, 2009; Hood et al., 2009; Fellman et al., 2010). Variability in nutrient (N, P, Si, Fe) sources across the wider NPCTR region, however, remains largely

unexplored. A concerted effort across the region is thus needed to better understand how this diversity of nutrient sources impact nutrient availability and subsequent autotrophic and heterotrophic production in coastal waters (Bidlack et al., 2017).

## Author contributions

K.S. conducted the data analyses and wrote the manuscript. B.H., S.T., I.G., A.O. and K.L. designed the study. I.G. coordinated

the routine freshwater sampling campaigns and data management. M.K. coordinated the storm event samplings and data management, with oversight from W.F.. W.F. and M.K. collected, processed and managed the discharge and weather data across the study watersheds. S.T., B.H., I.G., M.K., and A.O. designed the freshwater plume surveys, which were implemented by M.K. and A.O. All authors contributed to the drafting of the manuscript.



**Competing interests**

The authors declare no competing interests.

**Data availability**

Weather records, and nutrient and carbon chemistry data and fluxes are available in St. Pierre et al. (2020). Discharge data are available in Korver et al. (2019). Weather station data were compared against the downscaled climate normals for each station, extracted using station coordinates (Table S3) in the open access web service (climatewna.com, from Wang et al., 2016).

**Acknowledgements**

We gratefully acknowledge that this work was conducted on the traditional, ancestral and unceded territories of the Heiltsuk and Wuikinuxv First Nations. We would like to sincerely thank the Tula Foundation for funding this research, and the Hakai Institute for designing and operating the Kwakshua Watersheds Observatory, with special thanks to Jennifer Jackson for coordinating sampling campaigns and data management for the oceanography team. Field work was conducted by Bryn Fedje,
Lucy Quayle, Emma Myers, Christopher Coxson, Isabelle Desmarais, Emily Haughton, Chris Mackenzie, Skye McEwan, Chris O'Sullivan, and Rob White. Glen Woodsworth kindly provided the raw data associated with the Roddick 1983 report from the Geological Survey of Canada. Keith Holmes produced Fig. 1, and Frances Biles provided the shapefile delineation of the NPCTR regions used in Fig. 1. We also thank Paul Sanborn for useful discussion on the role of bedrock geomorphology and pedology in affecting freshwater nutrient exports, and Mark Garrison for producing Figure 8.

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





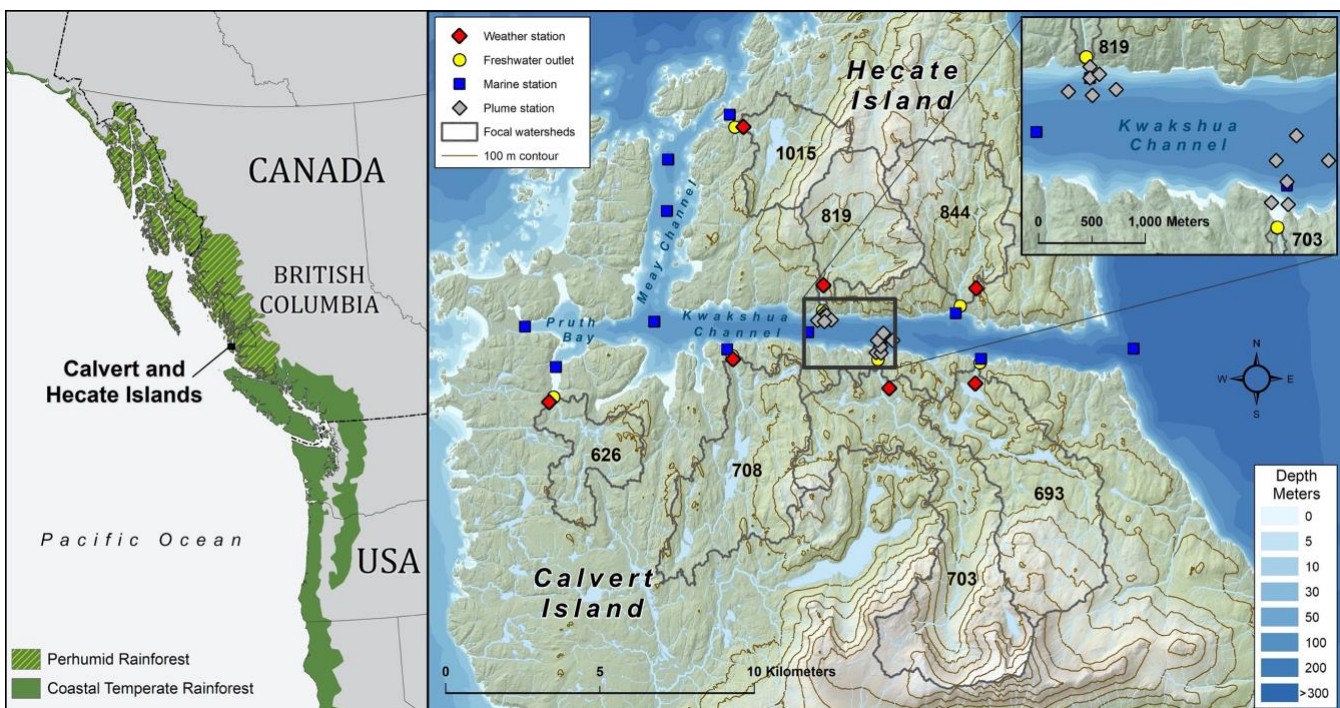

**Figure 1: Map of study area within the North Pacific coastal temperate rainforest region. All sampled freshwater (*n* = 7) and marine stations (*n* = 13) are shown. Inset indicates the sampling stations for the riverine plume surveys. The coastal temperate rainforest delineation is based on the detailed mapping by Ecotrust et al. (2015).**



**Figure 2: Climatology of Calvert and Hecate islands over the study period (01-Aug-2014 to 31-Dec-2018): (a) monthly mean, minimum and maximum air temperature across the seven watershed outlets; (b) mean monthly rainfall across the seven watershed outlets and total monthly discharge to Kwakshua Channel; (c) mean ± SE surface water temperature at the marine stations during sampling events, measured using an YSI; (d) Oceanic Niño Index, with El Niño and La Niña events highlighted based on thresholds of greater than ± 0.5 (weak event) or greater than ± 1.0 (strong event).**




**Figure 3: Time series of mean (± SE) measured monthly chlorophyll *a* (a), nutrient (b-d) and dissolved organic carbon (e) concentrations across both marine (across 0, 1, and 5 m depths) and freshwater stations. Freshwater fluxes to nearshore waters are also shown, upscaled from the measured fluxes to the entire drainage area of the Kwakshua Channel system. $NO_3^-$, combined nitrate-nitrite; $Si(OH)_4$, silicic acid; $PO_4^{3-}$, orthophosphate.**


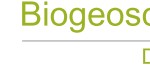
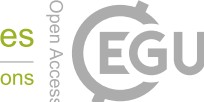

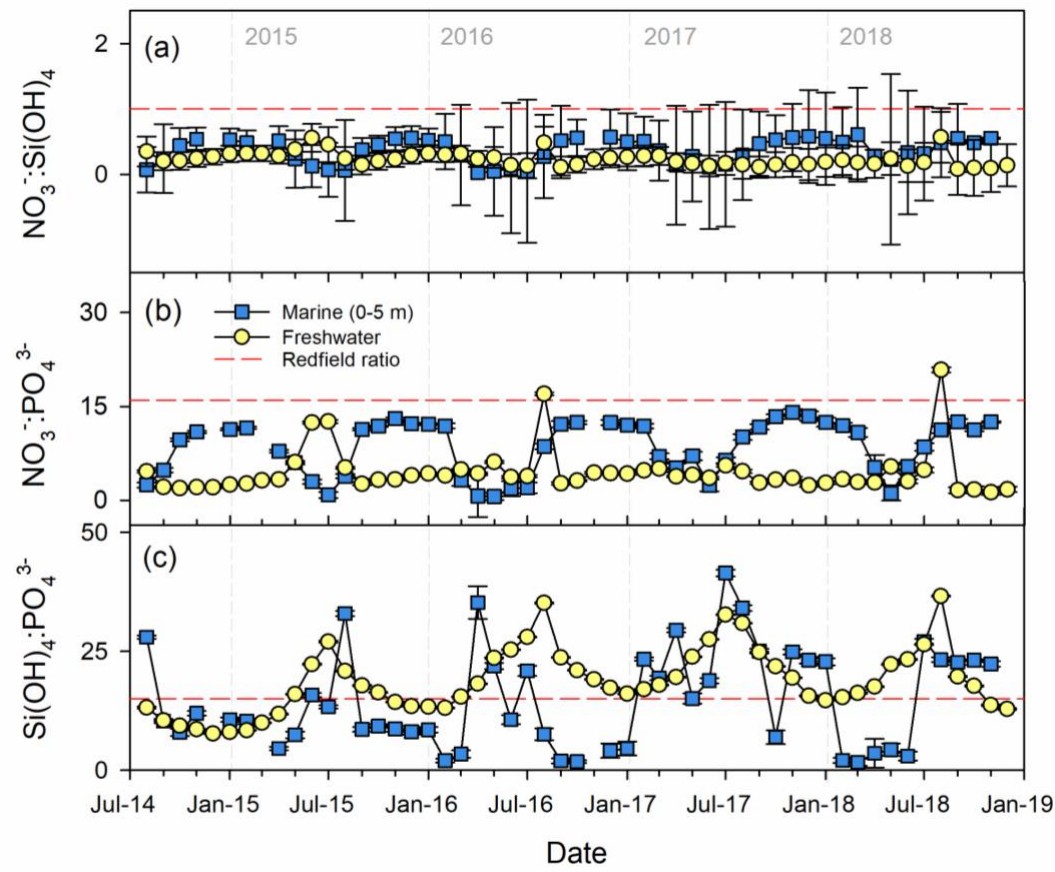

**Figure 4: Time series of monthly mean (± SE) stoichiometric ratios for NO$_3^-$: Si(OH)$_4$ (a), Si(OH)$_4$: PO$_4^{3-}$ (b), and NO$_3^-$: PO$_4^{3-}$ (c) across the marine (integrated 0-5 m) and freshwater stations. Freshwater stoichiometric ratios calculated from the nutrient fluxes (see Fig. 3). The extended Redfield ratio is shown as a point of reference (red dashed line).**






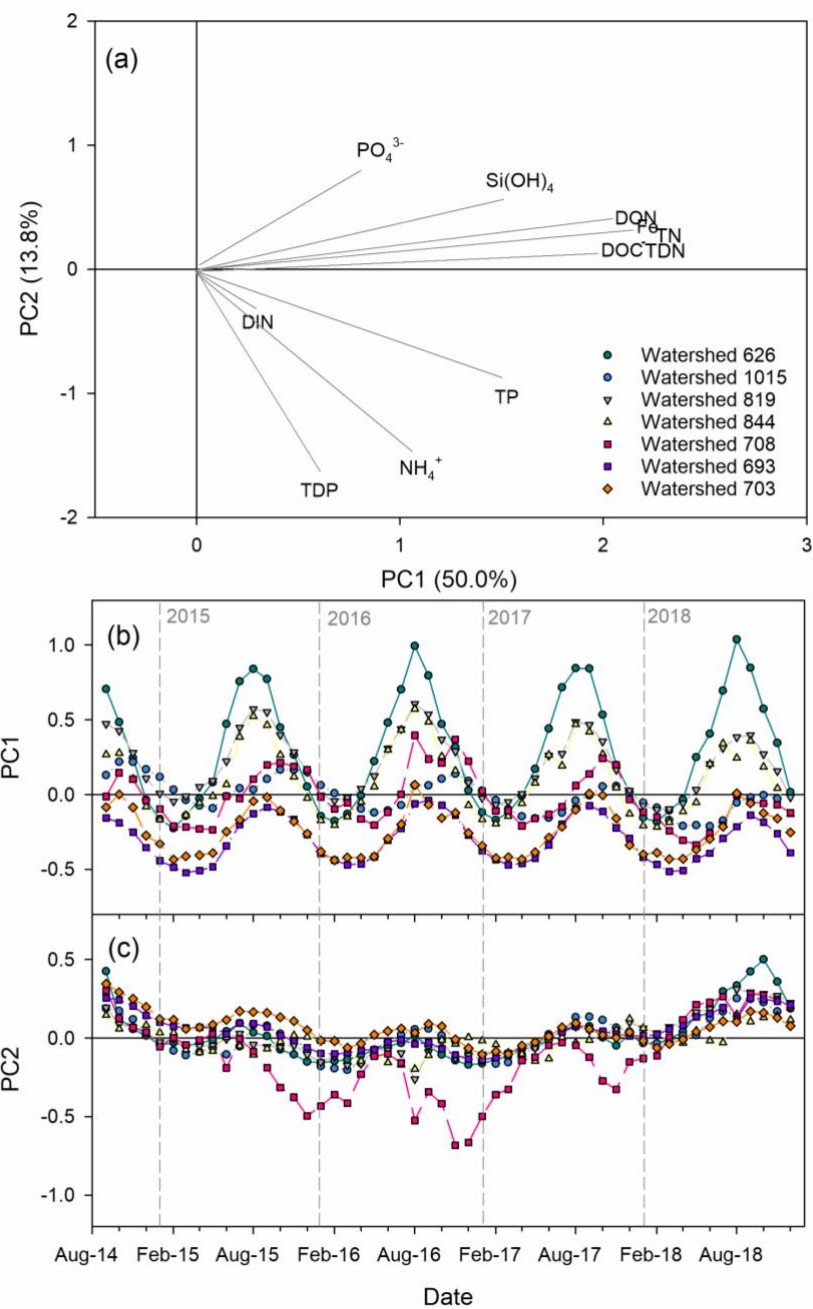

**Figure 5. Principal component analysis of flow-weighted bulk freshwater chemistry exports to the nearshore environment. (a) Nutrients in PC space (scaling 2 shown to preserve correlation between variables). Changes in PC1 (b) and PC2 (c) over time (scaling 1 – shown to preserve distance between objects). Note that PC1 loadings were multiplied by -1 to facilitate interpretation.**




**Figure 6: Nutrient concentration mixing plots across the freshwater plume (0, 1 and 5 m at six stations) at the outlet of watershed 819 for the 86.8 mm rainfall event on 7-Aug-2015 (day 5 of 6-day event). Combined nitrate-nitrite (NO₃⁻+NO₂⁻; panel a), orthophosphate (PO₄³⁻; panel b), silicic acid (Si(OH)₄; panel c), dissolved organic carbon (DOC; panel d), dissolved organic nitrogen (DON; panel e), and dissolved iron (dFe; panel f) are shown. Freshwater and marine end-members are the mean concentration at**
**watershed 819 for August 2015 and the plume sample with the highest salinity, respectively.**



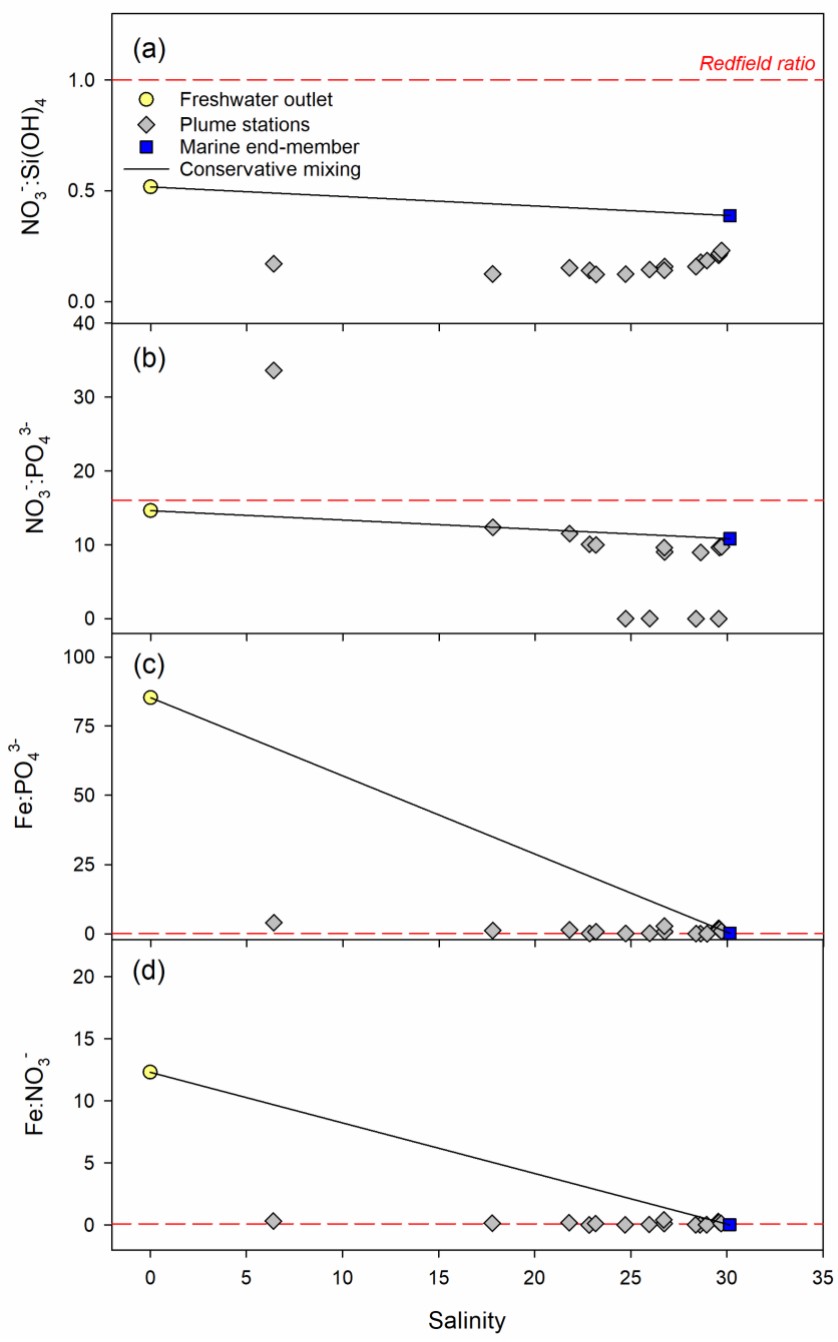

**Figure 7: Stoichiometric ratios across the freshwater plume (0, 1 and 5 m at six stations) at the outlet of watershed 819 during the rainfall event on 7-Aug-2015. Freshwater and marine end-members are the mean concentration at watershed 819 for the month of August and the plume sample with the highest salinity, respectively.**



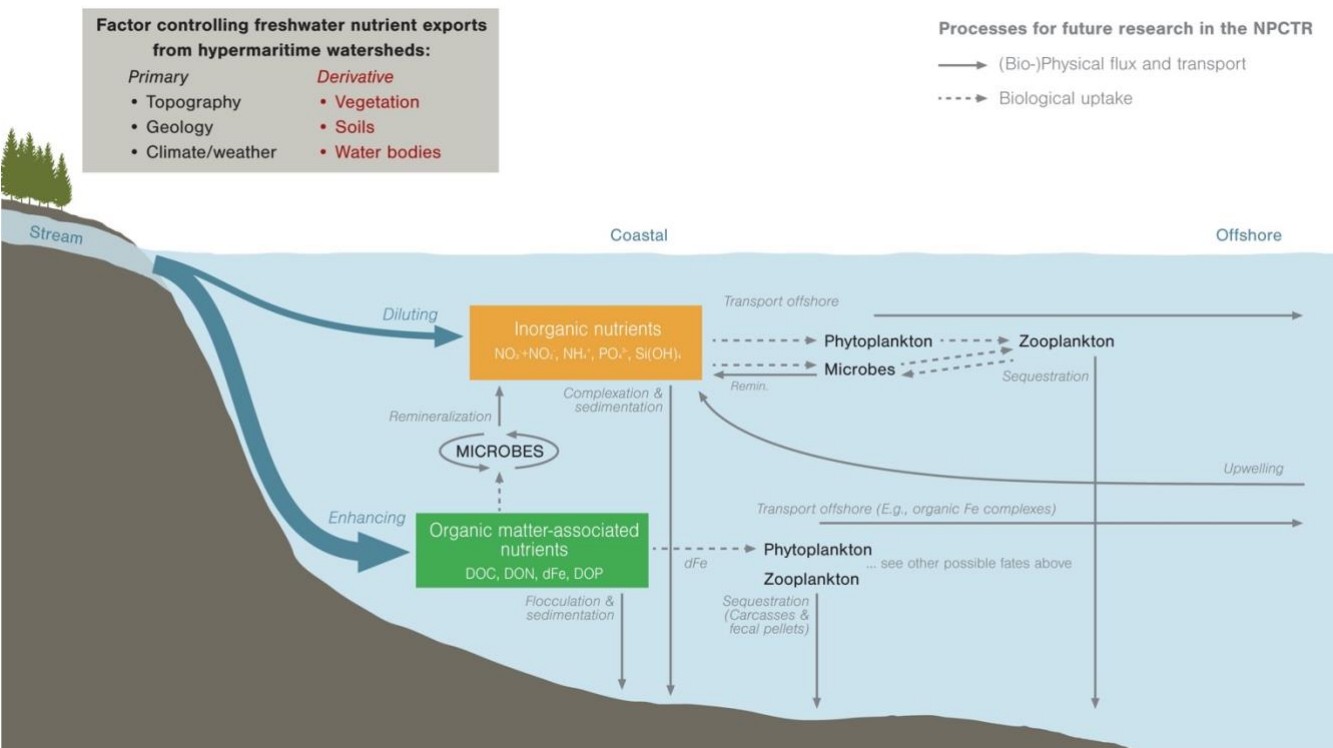


**Figure 8. Conceptual framework highlighting nutrient dynamics in coastal waters downstream of watersheds dominated by bog forests of the NPCTR. The size of arrows shows the relative magnitude of these processes (if known). Grey arrows indicate fluxes that are poorly quantified in this region.**



**Table 1. Annual combined fluxes of nutrients from the seven gauged watersheds (46.9 km²), upscaled to the drainage area of the Kwakshua Channel system (69.6 km²), and mean ± SE annual flow-weighted concentrations (C_F.W.).**

| Year | Flux (Mmol yr⁻¹) | | | | C_F.W. (μmol L⁻¹) | | | |
|---|---|---|---|---|---|---|---|---|
| | 2015 | 2016 | 2017 | 2018 | 2015 | 2016 | 2017 | 2018 |
| Water (km³) | 0.172 | 0.186 | 0.179 | 0.161 | - | - | - | - |
| DOC | 145±41 | 156±44 | 144±42 | 129±37 | 843±238 | 837±235 | 809±235 | 800±228 |
| TN | 2.06±0.59 | 2.30±0.65 | 2.18±0.63 | 1.88±0.54 | 11.98±3.43 | 12.37±3.51 | 12.18±3.52 | 11.69±3.33 |
| TDN | 1.83±0.44 | 2.09±0.50 | 1.96±0.48 | 1.64±0.39 | 10.64±2.56 | 11.23±2.67 | 10.96±2.66 | 10.18±2.45 |
| DON | 1.70±0.42 | 1.89±0.47 | 1.79±0.45 | 1.51±0.37 | 9.86±2.43 | 10.17±2.51 | 10.03±2.53 | 9.40±2.30 |
| DIN | 0.18±0.16 | 0.20±0.19 | 0.19±0.18 | 0.17±0.15 | 1.06±0.95 | 1.09±1.03 | 1.05±1.00 | 1.04±0.95 |
| $NH_4^+$ | 0.10±0.10 | 0.14±0.16 | 0.11±0.12 | 0.06±0.07 | 0.56±0.62 | 0.73±0.83 | 0.62±0.69 | 0.39±0.45 |
| [1]$NO_3^-+NO_2^-$ | 0.09±0.20 | 0.07±0.25 | 0.08±0.22 | 0.10±0.17 | 0.50±1.54 | 0.36±1.86 | 0.43±1.69 | 0.65±1.40 |
| $Si(OH)_4$ | 0.24±0.11 | 0.24±0.11 | 0.22±0.10 | 0.21±0.10 | 1.37±0.65 | 1.28±0.59 | 1.23±0.58 | 1.30±0.61 |
| TP | 0.07±0.04 | 0.08±0.05 | 0.07±0.04 | 0.06±0.04 | 0.39±0.24 | 0.42±0.25 | 0.40±0.24 | 0.38±0.23 |
| TDP | 0.05±0.03 | 0.07±0.04 | 0.06±0.03 | 0.04±0.02 | 0.30±0.16 | 0.37±0.20 | 0.34±0.18 | 0.26±0.14 |
| $PO_4^{3-}$ | 0.02±0.01 | 0.01±0.01 | 0.01±0.01 | 0.01±0.01 | 0.09±0.08 | 0.07±0.07 | 0.07±0.06 | 0.08±0.08 |
| Fe | 0.69±0.21 | 0.74±0.23 | 0.66±0.21 | 0.58±0.18 | 4.00±1.23 | 3.99±1.22 | 3.72±1.17 | 3.58±1.12 |
| *Key ratios* | | | | | | | | |
| $NO_3^-$: $Si(OH)_4$ | 0.29±0.05 | 0.22±0.05 | 0.18±0.05 | 0.16±0.09 | - | - | - | - |
| $NO_3^-$: $PO_4^{3-}$ | 4.24±0.05 | 4.57±0.05 | 3.86±0.06 | 2.97±0.12 | - | - | - | - |
| $Si(OH)_4$: $PO_4^{3-}$ | 14.47±0.04 | 20.20±0.03 | 21.64±0.03 | 18.50±0.04 | - | - | - | - |
| Fe:$PO_4^{3-}$ | 47.83±0.04 | 67.50±0.04 | 70.74±0.03 | 55.42±0.04 | - | - | - | - |

[1]$NO_3^-+NO_2^-$ fluxes and C_F.W. calculated as the difference between DIN and $NH_4^+$.





**Table 2. Mean (range) annual nutrient concentrations (in µmol L⁻¹) and stoichiometric ratios (mol mol⁻¹) across the 13 marine stations in Meay and Kwakshua Channels.**

| | 2015 | n | 2016 | n | 2017 | n | 2018 | n |
|---|---|---|---|---|---|---|---|---|
| *Concentrations* | | | | | | | | |
| DOC | 71.64 (7.87-965) | 364 | 86.85 (34.47-606) | 70 | 96.41 (63.95-131.28) | 28 | 105.82 (59.02-198.89) | 16 |
| $NO_3^- + NO_2^-$ | 8.10 (0.01-22.39) | 526 | 11.45 (0.01-20.35) | 136 | 7.86 (0.02-22.07) | 108 | 10.12 (0.02-19.64) | 34 |
| $Si(OH)_4$ | 19.17 (0.12-40.70) | 535 | 23.12 (1.30-34.34) | 135 | 16.84 (0.04-39.64) | 108 | 20.15 (0.11-33.90) | 33 |
| $PO_4^{3-}$ | 0.77 (<D.L.-2.50) | 523 | 1.01 (0.04-1.61) | 136 | 0.68 (<D.L.-1.76) | 106 | 0.90 (0.01-1.50) | 33 |
| *Ratios* | | | | | | | | |
| $NO_3^-$: $Si(OH)_4$ | 0.24 (<0.01-2.30) | 526 | 0.32 (<0.01-0.61) | 135 | 0.29 (<0.01-1.09) | 108 | 0.40 (0.08-4.01) | 33 |
| $NO_3^-$: $PO_4^{3-}$ | 5.95 (0.05-317) | 515 | 7.00 (0.06-14.01) | 136 | 7.87 (0.46-83.15) | 103 | 7.63 (0.55-39.10) | 33 |
| $Si(OH)_4$: $PO_4^{3-}$ | 11.19 (0.76-1110) | 525 | 6.92 (1.65-433) | 135 | 18.05 (1.67-568) | 103 | 8.04 (0.25-34.30) | 33 |