# Peer review of "Rain-fed streams dilute inorganic nutrients but subsidise organic matter-associated nutrients in coastal waters of the northeast Pacific Ocean"

_Biogeosciences, 2020_

## Referee Comment (RC1) · Anonymous Referee #1 · 13 Dec 2020

General comments

This paper presents valuable and unique riverine nutrient dataset with surprisingly low macronutrient concentrations. In general, we consider that riverine nutrient loadings fertilize coastal primary production and then ecosystems. However, the present study demonstrates a quite different picture. This paper contains useful data for both freshwater and marine researchers and can connect the separated research fields so far. The following points should be improved before this paper being considered for publishing in Biogeosciences.

[Figure]

Major comments

1. A link between Introduction-results-discussion-conclusion is not established well. In particular, the key issues described in Discussion section are not appropriately raised in Introduction section. Some improvements for this can lead the reader smoothly from Introduction to Conclusion.

2. Very low inorganic macronutrient concentrations in the present freshwater systems is unique and interesting. I would like to confirm whether freshwater nutrient concentrations in these watersheds have not been reported in past studies. If this is the first report, that should state clearly. If some previous studies exist, the authors should describe whether the present results are consistent with previous results. Another important message of this study may be the importance of measurements of organic nutrients such as DON and DOP in addition to inorganic macronutrients, because organic nutrients are less frequently analyzed for river waters.

Specific comments

3. L37. Silicon is usually treated as a macronutrient as shown by the authors for C:N:Si:P:Fe stoichiometory in the next paragraph.

4. L75. Readers need to know why this system should be targeted, but we cannnot find any explanations in Introduction section.

5. L215. Please show a reason for using the "Half".

6. L261. Fig. 2a –> 2b

7. L264. Highest temperatures are found in July from the X-axis label.

8. L269. From Fig.3a, highest Chl-a may be found in May 2015 for me.

9. L270. Detailed explanations are required how the authors determine the peaks of primary production.
10. L271. What is the advantage to show mean nutrient concentrations in marine waters regardless of such large seasonal variations?

11. L319. PN –> particulate N (PN = TN − TDN)

12. L387. St. Pierre and Oliver et al. –> St. Pierre et al.?

13. L390. The "assuming no loss" for dFe is generally not acceptable as shown in Fig. 6.

14. L413. So, what are these related to the Kwakshua Channel system?

15. L477. As well as heavily human-impacted watersheds, not impacted watersheds also probably have been studied. Such data can be picked up here and discussed with the present results.

16. L498. The terrestrial DON flux may elevate the contribution of riverine N on marine primary production of this system, but that is probably still quantitatively a small portion to support primary production.

17. L502. Kortzinger et al. 2001?

18. L508. The author should examine relative abundance of Si and N in subsurface waters, which is the primary source of nutrients for the Channel system surface waters. Si will be enriched than N in such source waters.

19. L631. Whether N input as DON have a significant impact on the spring bloom should be examined quantitatively with some assumptions. Lack of marine DON data may be an issue for this consideration.

———————————————

---

## Referee Comment (RC2) · Anonymous Referee #2 · 30 Dec 2020

General comments:

The paper presents an excellent and pertinent study of river/coast nutrients dynamics at two islands located at the Pacific eastern coast. The objectives are clearly stated and relevant and the methods are properly described. The presented data-sets obtained along four years of monitoring, in a relatively pristine environment, constitute also a relevant contribution of the presented work.

There are two main issues that will help to understand or further justify and explain

some aspects of the obtained results, and the authors could further comment or discuss: i) the monitored marine environment "Kwakshua Channel system" hydrodynamics and nutrients exchange with the Hakai passage that receives the contribution of much greater discharges from other watersheds are not sufficiently detailed; ii) the monitoring program concentrates on surface water and low attention is put on the sediment transport (together with associated nutrients) that should be more relevant at the bottom layers. The authors should further justify their option and comment on eventual consequences for their results.

Specific comments:

Line 62: "receives upwards of 2000 mm of rainfall per year". Is this an average maximum value? Please specify. Line 406 - Please, explicitly indicate the watersheds area instead of mention "smaller"

Technical corrections: Line 76: "(Giesbrecht et al. 2016 and in review " delete "and" Line 110: "(Gonzalez Arriola et al., 2015" – change to Arriola et al. 2015 Lines 159-160: The sentence is not clear. Please rephrase. Lines 253-254: Check or format the full range for temperature (use "to" instead of "-")

---

## Author Response (AR1)

March 19, 2021

**Re: Author response for** "**Rain-fed streams dilute inorganic nutrients but subsidise organic matter-associated nutrients in coastal waters of the northeast Pacific Ocean"**

Dear Dr. Suzuki,

Thank you very much for your positive assessment of our manuscript entitled, "Rain-fed streams dilute inorganic nutrients but subsidise organic matter-associated nutrients in coastal waters of the northeast Pacific Ocean". We would also like to sincerely thank the two reviewers for their thoughtful and considered comments, which have helped to improve the manuscript.

Please find attached our point-by-point responses, now including references to the lines in the manuscript where the changes can now be found. We have also included the revised manuscript, as well as a version of the revised manuscript where the changes are marked in blue.

Many thanks for your consideration of our manuscript for publication in Biogeosciences.

Best regards,

Kyra St. Pierre (on behalf of all co-authors)

**Point-by-Point Response**

**ANONYMOUS REFEREE #1**

**General comments**
This paper presents valuable and unique riverine nutrient dataset with surprisingly low macronutrient concentrations. In general, we consider that riverine nutrient loadings fertilize coastal primary production and then ecosystems. However, the present study demonstrates a quite different picture. This paper contains useful data for both freshwater and marine researchers and can connect the separated research fields so far. The following points should be improved before this paper being considered for publishing in Biogeosciences.

Thank you very much for your considered and constructive feedback!

**Major comments**
1. A link between Introduction-results-discussion-conclusion is not established well. In particular, the key issues described in Discussion section are not appropriately raised in Introduction section. Some improvements for this can lead the reader smoothly from Introduction to Conclusion.

Thank you very much for this constructive comment. We have critically examined the links between the manuscript sections and identified a couple of topics discussed later in the manuscript that were not introduced at the outset. Notably, this was the case for the impact of the El Niño Southern Oscillation on the land–ocean connection and variability in nearshore mixing, which are now raised within the Introduction section on lines 74-77 and 59-60, respectively.

2. Very low inorganic macronutrient concentrations in the present freshwater systems is unique and interesting. I would like to confirm whether freshwater nutrient concentrations in these watersheds have not been reported in past studies. If this is the first report, that should state clearly. If some previous studies exist, the authors should describe whether the present results are consistent with previous results.

Freshwater nutrient concentrations have not been previously reported for these watersheds. Only a subset (Aug. 2014 to Dec. 2016) of the dissolved organic carbon (DOC) concentrations have been previously published in Oliver et al. (2017) and St. Pierre and Oliver et al. (2020). A statement to this effect has been added to the methods (L179-181).

Another important message of this study may be the importance of measurements of organic nutrients such as DON and DOP in addition to inorganic macronutrients, because organic nutrients are less frequently analyzed for river waters.

The sentence, "In contrast to the field's prevailing focus on inorganic macronutrients ($NO_3^-$, $PO_4^{3-}$, $Si(OH)_4$), our results highlight the need to measure organic nutrient (DON, DOP) concentrations to fully understand the impact of freshwater exports on nearshore ecosystems." has been added to the conclusions (L683-685).

Specific comments

3. L37. Silicon is usually treated as a macronutrient as shown by the authors for C:N:Si:P:Fe stoichiometory in the next paragraph.

The word "micronutrients" has been exchanged for "other nutrients".

4. L75. Readers need to know why this system should be targeted, but we cannot find any explanations in Introduction section.

The rationale for using the North Pacific Coastal Temperate Rainforest (NPCTR) more broadly as a study region is highlighted in the preceding paragraph, namely that there is a strong connection between land and sea across the region. To reinforce this idea, we now state in the introduction, "The Calvert and Hecate Island systems are broadly representative of the many small rainfall-dominated coastal watersheds that define the outer coast of the NPCTR in BC and Alaska (Oliver et al., 2017). Because of the large freshwater fluxes from the NPCTR to the coastal ocean, it is an ideal region in which to examine the connection between land and sea (Bidlack et al., 2021)." (L81-84)

5. L215. Please show a reason for using the "Half".

Applying half the limit of detection is one of several possible conventions for dealing with analytical results below detection (Analytical Methods Committee, 2001). This approach is widely used, including by the National Pollutant Release Inventory of the Canadian federal government (link), and recognizes that values below this limit cannot be differentiated from any background noise, but is somewhat arbitrary (Croghan and Egeghy, 2003). While we acknowledge the biases of this method (Helsel, 2009), it is a reasonable approach for this dataset given the large number of parameters and systems that were modeled. We have added an explanation to this effect in the methods (L225-229).

Analytical Methods Committee: What should be done with results below the detection limit? Mentioning the unmentionable. amc technical brief, Royal Society of Chemistry (Ed.), Vol. 5, 2001.

Croghan, C. and Egeghy, P.P.: Methods of dealing with values below the limit of detection using SAS. Presented at Southeastern SAS User Group, St. Petersburg Florida, September 22-24, 2003.

Helsel, D.R.: Summing Nondetects: Incorporating Low-Level Contaminants in Risk Assessment. Integrated Environmental Assessment and Management, 6, 361-366, 2009.

6. L261. Fig. 2a –> 2b

Reference to Fig. 2a-c has now been made at the end of this sentence (now L272).

7. L264. Highest temperatures are found in July from the X-axis label.

The text has been changed to reflect this (now L275).

8. L269. From Fig. 3a, highest Chl-a may be found in May 2015 for me.

We have exchanged this sentence to reflect the entire range of chlorophyll a concentrations observed (<0.01 to 14.67 µg L$^{-1}$) - now L280. The seasonal cycle is described in the following sentence and the highest chlorophyll *a* concentrations can be observed at any time between April and June in any given year.

9. L270. Detailed explanations are required how the authors determine the peaks of primary production.

We recognize that this statement was ambiguous as the "peak" (i.e., highest mean monthly concentration of chlorophyll *a*) occurs in different months during each of the four study years: May in 2015 and 2017, June in 2016, and April in 2018. We have therefore edited the text to read, "… production peaked between April and June and again between July and August…" (L281). Peaks were identified as those time periods where concentrations of chlorophyll *a* were at their highest during the annual cycle.

10. L271. What is the advantage to show mean nutrient concentrations in marine waters regardless of such large seasonal variations?

The rationale behind showing mean nutrient concentrations in marine waters was to contrast these with the freshwater systems (i.e., L307-308). From these measurements, it is apparent that – on average – concentrations of $NO_3^-$, $PO_4^{3-}$, and $Si(OH)_4$ far exceed those in freshwaters, but freshwater concentrations of DOC far exceeded those in marine waters. Seasonal distinctions are discussed thereafter.

11. L319. PN –> particulate N (PN = TN - TDN)

The change has been made (now L330-331).

12. L387. St. Pierre and Oliver et al. –> St. Pierre et al.?

St. Pierre and Oliver shared first authorship of the 2020 publication in Limnology and Oceanography, hence why the reference is written as St. Pierre and Oliver et al. (2020) throughout the manuscript.

13. L390. The "assuming no loss" for dFe is generally not acceptable as shown in Fig. 6.

We completely agree with the reviewer that this is a major assumption. We are, however, lacking data on specific rates of dFe loss/reprocessing in nearshore waters of this region, so would be remiss to attempt an estimate otherwise. In acknowledgement of the limitations of this assumption, we have added a caveat in the sentence that follows this estimate: "However, given our observations (Fig. 6) and what is known about dFe loss from the water column in nearshore ecosystems, we acknowledge that the assumption of no loss is likely not realized and as such, these estimates represent a hypothetical upper bound on the possible stimulation of primary production by freshwater dFe exports." (L404-406)

14. L413. So, what are these related to the Kwakshua Channel system?

We have now added a statement in the discussion that applies these general findings about ENSO in the coastal northeast Pacific Ocean to our study region. Whereas Whitney and Welch

(2002) observed the differences in nutrient availability across the wider northeast Pacific region, we do not find this to be the case at the scale of the Kwakshua Channel system, where seasonal nutrient depletion and replenishment was fairly consistent between years, suggesting a stronger influence of local oceanographic processes. This conclusion has been added to section 4.1 (L429-431).

15. L477. As well as heavily human-impacted watersheds, not impacted watersheds also probably have been studied. Such data can be picked up here and discussed with the present results.

We searched the literature for examples of paired freshwater-marine nutrient surveys that were not within heavily human-impacted watersheds, but had difficulty coming up with examples where there was no significant human impact (logging, mining, agriculture, developments). There are some examples within the Arctic; however, Arctic watersheds are undergoing rapid environmental change (permafrost thaw, enhanced erosion, changing terrestrial productivity) that is significantly altering the exports of terrestrial materials and nutrients to nearshore environments such that comparing the Central Coast study sites to those in the Arctic is not without significant drawbacks. That being said, we have added an additional sentence to the paragraph further expanding on the reference to Cuevas et al., 2019 (L489-491).

Cuevas, L. A., Tapia, F. J., Iriarte, J. L., González, H. E., Silva, N., and Vargas, C. A.: Interplay between freshwater discharge and oceanic waters modulates phytoplankton size-structure in fjords and channel systems of the Chilean Patagonia, Progr. Oceanogr., 173, 103-113, https://doi.org/10.1016/j.pocean.2019.02.012, 2019.

16. L498. The terrestrial DON flux may elevate the contribution of riverine N on marine primary production of this system, but that is probably still quantitatively a small portion to support primary production.

Without site-specific regeneration rates, it is difficult to quantify (without a large degree of uncertainty) the impact that this DON flux has on marine primary production of the Kwakshua Channel system. The application of nitrogen regeneration potentials from other coastal areas to the Calvert and Hecate Island DON flux is the extent to which we feel comfortable quantitatively speculating on this potential subsidy, as described in more detail in response to point #19.

17. L502. Kortzinger et al. 2001?

The year "2001" has been added to the reference (now L526).

18. L508. The author should examine relative abundance of Si and N in subsurface waters, which is the primary source of nutrients for the Channel system surface waters. Si will be enriched than N in such source waters.

This is an excellent suggestion for future research within this system! Bottom water nutrient chemistry data have been collected from a small number of sites since the start of the sampling program, but has neither the spatial nor temporal resolution of the surface water dataset. As suggested by the other reviewer, we have expanded our discussion of the possible influence of bottom waters on surface nutrient biogeochemistry (L630-636).

19. L631. Whether N input as DON have a significant impact on the spring bloom should be examined quantitatively with some assumptions. Lack of marine DON data may be an issue for this consideration.

Earlier in this section (L505-510), we describe a quantitative assessment of the potential regeneration of DIN from DON using regeneration potentials from the Arctic region; however, these regeneration potentials are already speculative in that they are not based on local data, and we feel that extending these regeneration potentials to possible subsidy of the spring bloom would add too much uncertainty. We further describe the uncertainty of the initial estimate in the revised manuscript (L510-513). As stated by the reviewer, the lack of detailed marine DON and $NH_4^+$ data do hinder our ability to place bounds on these regeneration estimates. Understanding nitrogen dynamics in nearshore waters is a top priority for research moving forwards in this area.

**ANONYMOUS REFEREE #2**

**General comments:**
The paper presents an excellent and pertinent study of river/coast nutrients dynamics at two islands located at the Pacific eastern coast. The objectives are clearly stated and relevant and the methods are properly described. The presented data-sets obtained along four years of monitoring, in a relatively pristine environment, constitute also a relevant contribution of the presented work.

Many thanks for this positive assessment of our study!

There are two main issues that will help to understand or further justify and explain some aspects of the obtained results, and the authors could further comment or discuss: i) the monitored marine environment "Kwakshua Channel system" hydrodynamics and nutrients exchange with the Hakai passage that receives the contribution of much greater discharges from other watersheds are not sufficiently detailed; ii) the monitoring program concentrates on surface water and low attention is put on the sediment transport (together with associated nutrients) that should be more relevant at the bottom layers. The authors should further justify their option and comment on eventual consequences for their results.

i) The hydrodynamics of Kwakshua Channel are indeed influenced by Hakai Pass and Fitz Hugh Sound, both of which integrate exports from much larger watersheds along the coast; however, this impact is difficult to constrain as the logistics of sampling across the area as frequently as was done for the Kwakshua Channel system would be prohibitive. In acknowledgement of this influence, though, we have now stated in the site description (section 2.1): "Fitz Hugh Sound and Hakai Pass are also influenced by large freshwater fluxes from the British Columbia mainland, which affect marine nutrient dynamics within the system against which the watersheds are compared; however, quantifying the influence of these mainland freshwater exports is beyond the scope of this study." (L101-103)

ii) Surface waters (0-5 m) were the focus of the sampling program because these waters are the area of both highest freshwater influence and highest primary production. This is a valid point about the connection between surface and bottom waters with respect to nutrient dynamics along the coast. We unfortunately only have a very small number of deeper water samples

that do not match the spatial or temporal resolution of the surface samples, therefore limiting the conclusions about the connection between surface and bottom waters that can be drawn. Particulate organic carbon exports were previously reported for these watersheds (St. Pierre and Oliver et al., 2020) and are one to two orders of magnitude lower than for the dissolved organic carbon, suggesting that sediment export from these watersheds are of relatively little importance compared to the dissolved fluxes. The potential remobilisation of Fe from the sediments was discussed in the initial submission, but we have added a new paragraph to this discussion in section 4.2 to further elaborate on the connection between bottom waters and surface processes (L630-636).

**Specific comments**:

Line 62: "receives upwards of 2000 mm of rainfall per year". Is this an average maximum value? Please specify.

2000 mm per year is the median annual precipitation for the North Pacific coastal temperate rainforest region as described in Della Salla, 2011. Site-specific precipitation across the region can range between 1000 mm and more than 4000 mm, depending on location, elevation and year.

The sentence has been edited to read: "… receives on average 2000 mm of rainfall per year, with some locations receiving upwards of 4000 mm (Della Sala, 2011)." (now L63-64)

Line 406 - Please, explicitly indicate the watersheds area instead of mention "smaller"

In the following line (now L421), we now specify the range of the study watershed areas (3.2–12.8 km$^2$) as this is the range of watersheds over which we are comfortable that the decoupling from wider scale climate anomalies takes place.

**Technical corrections:**

Line 76: "(Giesbrecht et al. 2016 and in review " delete "and"

"and" has been exchanged for a comma.

Line 110: "(Gonzalez Arriola et al., 2015" – change to Arriola et al. 2015

Gonzalez is part of the researcher's surname and has not been removed from the citation.

Lines 159-160: The sentence is not clear. Please rephrase.

The subject of the second clause of the sentence seems to have been lost! The sentence now reads: "However, rainfall is a notoriously difficult climate parameter to measure accurately, let alone model and model estimates should improve over time with the incorporation of additional data sources." (now L167)

Lines 253-254: Check or format the full range for temperature (use "to" instead of "-")

Biogeosciences requests that the en dash "–" be used for specifying ranges. Per this guideline, we have exchanged the hyphen "-" for "–" throughout the manuscript.